# An Inexact Regularized Adaptive Algorithm with Manifold Identification for Training Structured Neural Networks

## Abstract

We propose an inexact regularized adaptive dual averaging algorithm with momentum, RAMDA, for training structured neural networks in various tasks by leveraging the help of regularization. Through the theory of manifold identification, we show that even in the presence of subproblem solution inexactness inevitable for adaptive methods with nonsmooth regularization, after a finite number of steps, the structure of the iterates generated by RAMDA are all identical to that induced by the regularizer at the stationary point of asymptotic convergence. This structure is locally optimal near the point of convergence and hence provides the best performance possible among all methods converging to the same point. Being able to produce stochastic gradient estimators converging almost surely to the true gradient even when the training problem is not a finite-sum but a stochastic one due to data augmentation, RAMDA is the first adaptive method with nonsmooth regularization to achieve this critical property for training structured deep learning models. With the simultaneous presence of a preconditioner and a regularization term, the subproblems of RAMDA as well as those of existing frameworks have no closed-form solutions, so we also propose a general iterative subroutine for approximately solving such subproblems efficiently while retaining similar guarantees for convergence and manifold identification. Extensive numerical experiments in modern computer vision, language processing, and speech tasks show that our subproblem solver is efficient and also applicable to existing frameworks, and the proposed RAMDA excels state of the art to generate deep learning models that are more structured without decreasing the prediction performance.

## 1 Introduction

Since the recent emergence of ChatGPT, large language models (LLMs) have gained much attention and popularity even among the public who are unfamiliar with machine learning. An issue of such gigantic neural network models is that they have hundreds of billions of model parameters, making their storage and inference expensive. It is thus important to find ways to exploit structures in the trained models to reduce their spatial and prediction costs without degrading the prediction performance. An active line of research is to explicitly use a regularizer in the training objective and apply proximal stochastic (sub)gradient methods to induce a desirable structure in the final model (Yang et al., 2019; Yun et al., 2021; Deleu & Bengio, 2021). However, it has been pointed out by Huang & Lee (2022) that these methods do not have any guarantees in finding a desirable structure due to the non-vanishing variance of the stochastic gradient estimators they use, and these algorithms indeed produce highly unstable and suboptimal structures empirically. Huang & Lee (2022) then proposed a regularized dual averaging method with momentum, abbreviated as RMDA, and leveraged the theory of manifold identification (Hare & Lewis, 2004; 2007; Lee, 2023) to show that models produced by RMDA can stably identify the locally optimal structure. Their

experiments demonstrated that their proposed method also empirically outperforms existing methods on modern computer vision tasks thanks to the ability of structure identification. Unfortunately, their method does not incorporate adaptiveness and thus might be only useful for computer vision related tasks.

For a wide range of tasks in deep learning such as language modeling and speech recognition, researchers have developed numerous architectures to achieve state-of-the-art prediction performance, including the popular transformer (Vaswani et al., 2017) and LSTM (Hochreiter & Schmidhuber, 1997). It is noteworthy that the transformer is also gaining prominence in computer vision (Liu et al., 2021), so it is becoming increasingly important to devise methods that attain satisfactory performance for training these network architectures with structure. For such modern architectures, adaptive algorithms like Adam (Kingma & Ba, 2015) that iteratively rescale the stochastic gradient update directions via a coordinate-wise preconditioner are known to outperform their non-adaptive counterparts (Denkowski & Neubig, 2017; Anil et al., 2019; Zhang et al., 2020; Liu et al., 2020; Kunstner et al., 2023). It is hence expected that the non-adaptive RMDA of Huang & Lee (2022) might not be quite promising for such architectures and tasks in language and speech, as well as the transformer models for computer vision, due to its lack of ability to learn some desired features faster while keeping the rest relatively steady like adaptive methods (Zhang et al., 2021).

This work aims to fill this gap to propose a regularized adaptive method that generates iterates identifying the active manifold, which represents all points possessing the structure identical to that at the point of convergence induced by the regularization. This structure is locally optimal in the sense that for including a sequence converging to the same point, the active manifold is of the lowest rank possible among the manifold collection of the structure class induced by the regularizer. A manifold of a lower rank means the model is more structured and thus the active manifold is desirable. Indeed, consider a simple example of $\ell_1$-norm regularization that promotes sparsity, whose associated manifold collection is the subspaces of different sparsity patterns, and assume that the point of convergence is $x^* = (0, 1, 2)$. Near $x^*$, $\|x^*\|_1$ is smooth along the two-dimensional active manifold $S :=$ $\{(0, a, 1) \mid a, b \in \mathbb{R}\}$. For any sequence $\{x^t\}$ converging to $x^*$, it is impossible that $\|x^t\|_0 < 2$ for all $t$ large, as we otherwise get $\|x^t - x^*\| \geq 1$. On the other hand, consider $x^t :=$ $(t^{-1}, 1 + t^{-1}, 2)$ and $y^t := (0, 1 + t^{-1}, 2 + t^{-1})$. Clearly, both sequences converge to $x^*$, but $x^t \notin S$ and $y^t \in S$ for all $t$, meaning that $\{y^t\}$ identifies the active manifold to possess desirable structure at the limit point $x^*$ but $\{x^t\}$ fails so. Our goal is to devise an algorithm with *adaptiveness* for training structured neural networks that produces iterates like $\{y^t\}$ above. Such a property for iterates not leaving the active manifold of the objective function is called manifold identification in nonlinear optimization (Hare & Lewis, 2007).

Given the pervasive usage of data augmentation in deep learning, we consider the case in which the objective function is the expectation over a probability distribution as follows.

$$\min_{W \in \mathcal{E}} \quad F(W) := \mathbb{E}_{\xi \sim \mathcal{D}}\left[f_\xi(W)\right] + \psi(W), \tag{1}$$

where $\mathcal{E}$ is a Euclidean space with inner product $\langle \cdot, \cdot \rangle$ and the induced norm $\|\cdot\|$, $\mathcal{D}$ is a distribution over a space $\Omega$ that represents all possible data modifications, $f_\xi$ is differentiable almost everywhere for all $\xi \in \Omega$, and the possibly nonsmooth term $\psi(W)$ is a regularizer for promoting a desirable structure in the optimal solutions. As discussed by Poon et al. (2018); Huang & Lee (2022), to achieve manifold identification for stochastic methods, it is necessary to drive the variance of the stochastic estimator of the gradient of the loss term to zero. Our method thus draws inspirations from Huang & Lee (2022) to consider a dual averaging approach to asymptotically reduce the variance of zero, but we go beyond their non-adaptive algorithm to incorporate an adaptive preconditioner in order to achieve better model predictive ability. Our algorithm also borrows ideas from MADGRAD of Defazio & Jelassi (2022) that combined adaptiveness, momentum, and dual averaging for unregularized training objectives, and our method can also be seen as a generalization of theirs to the regularized setting. These features result in our Regularized Adaptive Momentumized Dual Averaging (RAMDA) method. We will prove that even with adaptiveness and momentum added, RAMDA still attains variance reduction and convergence to a stationary point for the regularized training problem (1). Although Defazio & Jelassi (2022) provided some convergence guarantees for MADGRAD that can be seen as the special case of RAMDA when

$\psi \equiv 0$, their analysis is limited to the convergence rate of the objective value in convex problems. Our analysis of convergence in the nonconvex case, variance reduction, and the manifold identification property are new and closer to the properties desirable in practice. On the empirical side, our method achieves performance better than state of the art for training structured models on representative modern deep learning tasks.

Moreover, the major challenge of designing regularized adaptive methods is solving the subproblem. When adaptiveness is not involved, subproblems of existing methods for regularized training like RMDA have closed-form solutions through the proximal operator associated with the regularizer. Similarly, when no regularization is involved, the subproblems of adaptive methods like Adam or MADGRAD are smooth quadratic, so closed-form solutions can also be obtained efficiently. However, when a preconditioner for adaptiveness and a nonsmooth regularization term appears simultaneously, these subproblems no longer possess easily-computable closed-form solutions except for few special cases. To deal with this issue, for the subproblems of both our RAMDA and existing regularized adaptive methods, we propose an implementable inexact condition in the subproblem solving and a companion subproblem solver that efficiently compute approximate solutions satisfying this condition. As the earlier toy example has demonstrated, approximate subproblem solutions can easily fail manifold identification, but our analysis shows that our inexactness condition still ensures manifold identification and provides convergence guarantees to stationary points. When applied to existing methods, our inexactness condition is applicable to general regularizers, and it provides strong convergence guarantees similar to those in existing analyses for the exact versions.

Our major contributions are as follows.

1. **An adaptive algorithm for finding the locally optimal structure**: RAMDA is the first regularized adaptive method guaranteed to find the locally optimal structure possessed by the stationary point to which its iterates converge. It thus produces models that are more structured without decreasing the prediction performance. In comparison to RMDA that also guarantees structure identification, RAMDA is an adaptive one and thus provides better performance on a wide range of modern deep learning models as evidenced by our extensive experiments.

2. **Efficient subproblem solver for regularized adaptive methods**: We propose an implementable inexactness condition a companion efficient subproblem solver for approximate solutions of subproblems of regularized adaptive methods, including ours and existing ones, that have no closed-form solution. *Without additional assumptions on the problem class*, we show that the induced inexactness does not affect convergence guarantees or manifold identification. This condition and subproblem solver therefore also serve as the key step for realizing existing frameworks for regularized adaptive algorithms.

3. **A method with outstanding performance in practice**: Experiments on training modern neural networks in computer vision (ImageNet), language processing (Transformer-XL), and speech (Tacotron2) with structured sparsity show that RAMDA steadily outperforms state of the art to obtain higher structured sparsity ratio and better prediction performance at the same time.

## 2   ALGORITHM

Our algorithm can be seen as a dual averaging method that incorporates a proximal operation for the regularization, momentum, and an adaptive feature that computes a diagonal preconditioner from a weighted average of the squared norm of historical stochastic gradients. For the ease of the description, we assume without loss of generality that $\mathcal{E} = \mathbb{R}^n$ in this section. At the $t$th iteration, we first draw an independent and identically distributed sample $\xi_t \sim \mathcal{D}$, calculate $s_t := \eta_t \sqrt{t}$, compute the stochastic (sub)gradient $G^t := \nabla f_{\xi_t}(W^{t-1})$ of the loss function at the current point $W^{t-1}$ with respect to $\xi_t$, and then update the weighted sum $V_t$ of historical stochastic gradients and the weighted sum $U_t$ of their squared norms using the stepsize $s_t$:

$$\begin{cases} V_0 & := 0, \quad V_t := \sum_{k=1}^{t} s_k G^k = V_{t-1} + s_t G^t, \\ U_0 & := 0, \quad U_t := \sum_{k=1}^{t} s_k G^k \circ G^k = U_{t-1} + s_t G^t \circ G^t, \end{cases} \quad \forall t > 0, \quad (2)$$

**Algorithm 1:** RAMDA $(W^0, T, T_2, \epsilon, \{\eta_t\}, \{c_t\}, \{\epsilon_t\})$

---

$V^0 \leftarrow 0, \quad U^0 \leftarrow 0, \quad \alpha_0 \leftarrow 0$
**for** $t = 1, \ldots, T$ **do**
$\quad$ Sample $\xi_t \sim \mathcal{D}, \quad s_t \leftarrow \eta_t \sqrt{t}, \quad \alpha_t \leftarrow \alpha_{t-1} + s_t$
$\quad$ $G^t \leftarrow \nabla f_{\xi_t}(W^{t-1})$
$\quad$ Compute $V^t, U^t$ by (2), construct $P^t$ by (3), and $\theta_t \leftarrow \max(\text{diag}(P^t))^{-1}$
$\quad$ Compute $\tilde{W}^t$ in (4) by PG$(W^t, W^0, \alpha_t^{-1} V^t, \alpha_t^{-1} P^t, \alpha_t \theta_t, T_2)$
$\quad$ Update $W^t$ by (5)
**output:** $W^T$

---

where $\circ$ denotes the Hadamard (pointwise) product in $\mathcal{E}$. We then construct the preconditioner $P^t$ by

$$P^t := \text{Diag}(\sqrt[3]{U^t} + \epsilon), \tag{3}$$

where $\epsilon > 0$ is a (usually small) constant for numerical stability, and $\text{Diag}(\cdot)$ is the diagonal matrix whose diagonal entries are the elements of the input vector. The update direction is then obtained by (approximately) solving the following subproblem.

$$\tilde{W}^t \approx \arg\min_W \left( Q_t(W) := \langle V^t, W \rangle + \frac{1}{2} \langle W - W^0, P^t(W - W^0) \rangle + \alpha_t \psi(W) \right), \ \alpha_t := \sum_{k=1}^t s_k. \tag{4}$$

Details regarding the subproblem (4) and how to solve it are deferred to Section 3. Finally, the iterate is updated by averaging $\tilde{W}^t$ and $W^{t-1}$ using a prespecified factor $c_t \in [0, 1]$:

$$W^t = (1 - c_t) W^{t-1} + c_t \tilde{W}^t = W^{t-1} + c_t \left( \tilde{W}^t - W^{t-1} \right). \tag{5}$$

If one sets $P^t = \sqrt{t}I$, where $I$ is the identity matrix, this framework recovers RMDA of Huang & Lee (2022). On the other hand, with (3), adaptiveness is incorporated to obtain our RAMDA. The averaging step in (5) with $c_t \neq 1$ can be interpreted as incorporating a momentum term to compute the next iterate in the non-regularized non-adaptive case (Jelassi & Defazio, 2020; Tao et al., 2018). We also note that when $\psi \equiv 0$, RAMDA reduces to MADGRAD of Defazio & Jelassi (2022).

## 3 SUBPROBLEM SOLVER

Given the current iterate $W^t$, a momentum term $m_t$, a preconditioner $P^t$, and a stepsize $\eta_t$, existing regularized adaptive stochastic gradient algorithms for (1) can be summarized in the following general form (Yun et al., 2021):

$$W^t = \arg\min_W \left( \hat{Q}_t(W) := \langle m_t, W \rangle + \frac{1}{2\eta_t} \langle W - W^{t-1}, P^t(W - W^{t-1}) \rangle + \psi(W) \right), \tag{6}$$

which is very similar to (4). When the preconditioner $P^t$ is a multiple of the identity matrix like in the case of Huang & Lee (2022), the exact subproblem solution of (4) can be efficiently computed through the proximal operator associated with the regularizer. However, a major difficulty for realizing regularized adaptive methods, including the proposed RAMDA and the framework of Yun et al. (2021), whose preconditioners are not a multiple of the identity, is that except for few special regularizers, the subproblem usually has no closed-form solution. We therefore consider using approximate solutions of the subproblem in (4).

We propose to apply a few iterations of proximal gradient (PG) (see, *e.g.*, Beck & Teboulle, 2009; Nesterov, 2013) to approximately solve the subproblems in (4) and (6) when no closed-form solution is available, and we will show theoretically and empirically in the following sections that such approximate solutions have only little affects on the theoretical guarantees and the final model quality. For the inexactness of the approximate solution in (4), we require

$$\epsilon_t \geq \min_{s \in \partial Q_t(\tilde{W}^t)} \|s\|, \quad Q_t(\tilde{W}^t) \leq Q_t(W^{t-1}), \tag{7}$$

for some pre-specified $\epsilon_t$, where $\partial Q_t(W^{t+1})$ is the (limiting) subdifferential (see, *e.g.*, Rockafellar & Wets, 2009, Definition 8.3). This condition can be easily checked as the PG iterations would produce an element of $\partial \psi(W^{t+1})$, and the differential of the smooth part of $Q_t$ is straightforward. For the sake of time efficiency, we also impose an upper limit for the number of PG iterations. Likewise, when we apply our subproblem solver to (6), we assume (7) but with $Q_t$ replaced by $\hat{Q}_t$ and $\tilde{W}^t$ by $W^t$. We focus on the case that $P^t$ is diagonal, and thus the reciprocal of the largest eigenvalue $\max(\mathrm{diag}(P^t))$, where $\mathrm{diag}(\cdot)$ is the vector formed by the diagonal entries of the input matrix, can be calculated easily and be used to compute a step size guaranteeing sufficient objective decrease. For cases in which this value is difficult to obtain, one can apply a simple backtracking linesearch for the subproblem to find a suitable step size efficiently. This PG subproblem solver is summarized in Algorithm 2. To guarantee convergence for both our algorithm and the framework of Yun et al. (2021), our analysis in Section 4 requires that $\{\epsilon_t\}$ satisfy

$$\sum_{t=0}^{\infty} \epsilon_t^2 < \infty. \tag{8}$$

For any given $\epsilon_t$, we will show in Section 4 that (7) can be satisfied by our PG solver after a certain number of iterations.

---

**Algorithm 2:** PG $(Z^0, W^0, V, P, \theta, T_2, \hat{\epsilon})$

---

**if** $\psi$ *is nonconvex* **then** $\theta \leftarrow \theta/2$
**for** $j = 1, \dots, T_2$ **do**
    $Z^j \leftarrow \mathrm{prox}_\psi(Z^{j-1} - \theta(V + P(Z^{j-1} - W^0)))$
    **if** (7) *holds with $\epsilon_t = \hat{\epsilon}$ and $\tilde{W}^t = Z^j$* **then** $Z^{T_2} \leftarrow Z^j$ and break
**output:** $Z^{T_2}$

---

## 4 ANALYSIS

This section discusses theoretical guarantees for RAMDA and the proposed subproblem solver in Algorithm 2. We also prove convergence guarantees for applying PG to approximately solve (6) for the framework of Yun et al. (2021). Due to the space limit, all proofs are in the appendices. Some of our results are inspired by Huang & Lee (2022), but we note that with the added inexactness in (4) and the adaptiveness for the preconditioner, the analysis is nontrivial. Recall that we assume that $f_\xi$ is differentiable only almost everywhere but not everywhere, which conforms with widely used network structures like the ReLU-type activations.

We first show that (7) can be attained by our PG subproblem solver.

**Theorem 1.** *Assume that* (4) *and* (6) *has at least one optimal solution, with the optimal function value being finite. Given any $\epsilon_t > 0$, the number of iterations of Algorithm 2 needed for satisfying* (7) *for both* (4) *and* (6) *is $O(\epsilon_t^{-1})$ when $\psi$ is convex and $O(\epsilon_t^{-2})$ when $\psi$ is nonconvex.*

Next, we argue that the point of convergence $W^*$ of RAMDA is almost surely a stationary point such that $0 \in \partial F(W^*)$.

**Theorem 2.** *Consider $\{\tilde{W}^t\}$ generated by Algorithm 1 with* (7) *for* (1), *with $\{c_t\}$ and $\{\epsilon_t\}$ satisfying $\sum c_t = \infty$ and* (8). *Assume for any $\xi \sim \mathcal{D}$, $f_\xi$ is $L$-Lipschitz-continuously-differentiable almost surely for some $L$, so the expectation is also $L$-Lipschitz-continuously-differentiable, there is $C \geq 0$ such that $\mathbb{E}_{\xi_t \sim \mathcal{D}} \left\| \nabla f_{\xi_t}\left(W^{t-1}\right) \right\|^4 \leq C$ for all $t$, and that the set of stationary points $\mathcal{Z} \coloneqq \{W \mid 0 \in \partial F(W)\}$ is nonempty. For any given $W^0$, consider the event that $\{\tilde{W}^t\}$ converges to a point $W^*$ (each event corresponds to a different $W^*$). If $\partial \psi$ is outer semicontinuous at $W^*$, this event has a nonzero probability, and $\{\eta_t\}$ satisfy*

$$\sum s_t \alpha_t^{-1} = \infty, \quad \sum \left(s_t \alpha_t^{-1}\right)^2 < \infty, \quad \left\| W^{t+1} - W^t \right\| \left(s_t \alpha_t^{-1}\right)^{-1} \xrightarrow{a.s.} 0,$$

*then we have that $W^* \in \mathcal{Z}$ with probability one conditional on this event. Moreover, $\{W^t\}$ also converges to this stationary point $W^*$.*

Usually, convergence to a point requires some further regularity conditions like the Kurdyka–Łojasiewicz condition and boundedness of the iterates. However, existing frameworks regarding iterates convergence using such conditions also require the method analyzed to have a subgradient-descent-like behavior and to be a descent algorithm. Neither of these hold true even for the basic stochastic gradient algorithm, and we leave the analysis for this part as a challenging future work.

Our next key result shows that after a finite number of iterations, the iterates of RAMDA all possess the same structure as that of the point of convergence $W^*$. For this end, we first need to introduce the notions of partial smoothness and prox-regularity, and impose these assumptions on $\psi$ at $W^*$.

**Definition 1** (Partial Smoothness (Lewis, 2002; Hare & Lewis, 2004)). *A function $\psi$ is partly smooth at a point $W^*$ relative to a set $\mathcal{M}_{W^*} \ni W^*$ if*

*1. Around $W^*$, $\mathcal{M}_{W^*}$ is a $\mathcal{C}^2$-manifold and $\psi|_{\mathcal{M}_{W^*}}$ is $\mathcal{C}^2$.*
*2. $\psi$ is regular (finite with the Fréchet subdifferential coincides with the limiting Fréchet subdifferential) at all points $W \in \mathcal{M}_{W^*}$ around $W^*$ with $\partial\psi(W) \neq \emptyset$.*
*3. The affine span of $\partial\psi(W^*)$ is a translate of the normal space to $\mathcal{M}_{W^*}$ at $W^*$.*
*4. $\partial\psi$ is continuous at $W^*$ relative to $\mathcal{M}_{W^*}$.*

We often call $\mathcal{M}_{W^*}$ the active manifold at $W^*$. Locally, this manifold represents all points near $W^*$ that share the same structure as $W^*$ induced by the regularizer. Therefore, finding the active manifold is equivalent to finding the locally optimal structure.

**Definition 2** (Prox-regularity (Poliquin & Rockafellar, 1996)). *A function $\psi$ is prox-regular at $W^*$ for $V^* \in \partial\psi(W^*)$ if $\psi$ is locally lower semi-continuous around $W^*$, finite at $W^*$, and there is $\rho > 0$ such that $\psi(W_1) \geq \psi(W_2) + \langle V, W_1 - W_2 \rangle - \frac{\rho}{2}\|W_1 - W_2\|^2$ for every $W_1, W_2$ near $W^*$ with $\psi(W_2)$ close to $\psi(W^*)$ and $V \in \partial\psi(W_2)$ close to $V^*$. $\psi$ is prox-regular at $W^*$ if it is prox-regular for all $V \in \partial\psi(W^*)$.*

**Theorem 3.** *Consider Algorithm 1 with the conditions in Theorem 2 satisfied. Consider the event of $\{\tilde{W}^t\}$ converging to a certain point $W^*$ as in Theorem 2, if the probability of this event is nonzero; $\psi$ is prox-regular and subdifferentially continuous at $W^*$ and partly smooth at $W^*$ relative to the active $\mathcal{C}^2$ manifold $\mathcal{M}_{W^*}$; $\partial\psi$ is outer semicontinuous at $W^*$; and the nondegeneracy condition*

$$-\nabla f(W^*) \in \mathrm{relint}\, \partial\psi(W^*)$$

*holds at $W^*$, then conditional on this event, almost surely there is $T_0 \geq 0$ such that*

$$\tilde{W}^t \in \mathcal{M}_{W^*}, \quad \forall t \geq T_0.$$

*In other words, the active manifold at $W^*$ is identified by the iterates of Algorithm 1 after a finite number of iterations almost surely.*

We note particularly that convex and weakly-convex (Nurminskii, 1973) functions are all regular, prox-regular, and subdifferentially continuous everywhere.

We also show that the subproblem solver proposed in Section 3 can be effectively applied to the general framework of Yun et al. (2021) while still maintaining the same convergence guarantees. We note that our result is much stronger than that of Deleu & Bengio (2021) because we do not make any assumption on the regularizer, while their analysis for the inexact case has a very restrictive requirement such that the regularization term $\psi$ needs to be Lipschitz-continuously-differentiable, which excludes numerous widely-used regularizers in machine learning for inducing structures. Moreover, our inexact condition is easily checkable, while theirs for the objective distance to the optimum of the subproblem cannot be implemented.

**Theorem 4.** *For the framework in Yun et al. (2021) with the subproblem solved approximately by Algorithm 2 and that (7) holds with $\{\epsilon_t\}$ satisfying (8). Then Theorem 1 of Yun et al. (2021) still holds, but with the constants $\{Q_i\}$ being also dependent on $\sum_{t=0}^{\infty} \epsilon_t^2$.*

## 5 EXPERIMENTS

Following Wen et al. (2016), we consider structured sparsity in our experiments for training structured neural networks. We in particular employ the group lasso regularization (Yuan & Lin, 2006) to encourage group sparsity. We begin from demonstrating the efficiency and effectiveness of the PG solver described in Section 3 for both RAMDA and existing regularized adaptive methods. We then consider tasks in computer vision, language processing, and speech, and compare the following algorithms using Pytorch (Paszke et al., 2019).

- RAMDA: The proposed Algorithm 1.
- RMDA (Huang & Lee, 2022): A regularized modernized (non-adaptive) dual averaging method.
- ProxSGD (Yang et al., 2019): A non-adaptive proximal SGD method with momentum.
- ProxGen (Yun et al., 2021): A general framework of stochastic proximal methods. We follow their experimental setting to use an AdamW (Loshchilov & Hutter, 2019) implementation with regularization. The subproblem solver is Algorithm 2.
- ProxSSI (Deleu & Bengio, 2021): an implementation of ProxGen specifically designed for group sparsity. In particular, ProxSSI uses the Newton-Raphson algorithm to solve the subproblem to near optimality.

For each task, we also provide a dense baseline that did not include a group lasso regularizer in the training for reference, but our comparison is only among those methods for training structured models. The baseline is SGD with momentum (MSGD) for computer vision, and AdamW for the other two. These algorithms are summarized in Table 1. Our code for reproducing the experiments and the hyperparameter settings in the experiments are all available in the supplementary materials. Additional details on the stability in the structure (namely the level of structured sparsity here) over epochs of RAMDA is put in Appendix C.

Table 1: Algorithms used in the experiments.

| Algorithm | Unregularized counterpart | Subproblem |
|---|---|---|
| RAMDA | MADGRAD (Defazio & Jelassi, 2022) | PG |
| RMDA | MDA (Jelassi & Defazio, 2020) | Closed-form solution |
| ProxSGD | MSGD | Closed-form solution |
| ProxGen | AdamW (Loshchilov & Hutter, 2019) | PG |
| ProxSSI | AdamW (Loshchilov & Hutter, 2019) | Newton-Raphson |

Because we disable weight decay in these experiments, (regularized) AdamW actually reduces to (regularized) Adam. We use two criteria to compare the algorithms: 1. Model predictive ability, and 2. Structured sparsity level. The definition of model predictive ability depends on the specific problem, so we provide precise specifications for each problem in their respective sections. Regarding the structured sparsity, sparsifying deep neural networks while preserving its performance requires prior knowledge of model design. A commonly used approach is retaining certain parameters during the training process, and we adhere to this convention such that the bias, batch normalization (Ioffe & Szegedy, 2015), layer normalization (Ba et al., 2016), and embedding layers do not have any sparsity-inducing regularization imposed on them (Deleu & Bengio, 2021; Peste et al., 2021). For the rest, we adopt a channel-wise grouping for convolutional layers, an input-wise grouping for fully-connected layers, and also an input-wise grouping for LSTM layers during training phase. For evaluation, our structured sparsity is calculated using the weighted group sparsity with the weights proportional to the number of parameters in each group. This criterion better reflects the percentage of model condensation and inference acceleration.

Following Huang & Lee (2022), we introduce a restarting strategy to the implementation of RAMDA. At each stage of the training, the learning rate $\eta_t$ and the momentum factor $c_t$ are fixed throughout the stage. Once the epoch count enters the next stage, we reset the counter $t$ to 1 and use the output parameters $W^T$ from the previous round as the new input parameters $W^0$ to the same algorithm, set $\alpha_t, V^t$ and $U^t$ to 0, but keep the scheduling for $\eta$ and $c$ going without resetting them. The hyperparameter momentum value $c_t$ is initialized as either 0.1 or 0.01, depending on the problems. However, different from the suggestion of

Table 2: Group sparsity and validation accuracy of different subproblem stopping criteria.

| | No early stopping | | Early stopping | |
|---|---|---|---|---|
| Algorithm | Validation acc. | Group sparsity | Validation acc. | Group sparsity |
| Logistic regression/MNIST | | | | |
| ProxGen | 91.31% | 39.92% | 91.31% | 39.92% |
| RAMDA | 91.35% | 57.40% | 91.35% | 57.40% |
| VGG19/CIFAR10 | | | | |
| ProxGen | $92.70 \pm 0.22\%$ | $88.79 \pm 0.03\%$ | $92.67 \pm 0.11\%$ | $86.92 \pm 0.38\%$ |
| RAMDA | $92.70 \pm 0.21\%$ | $86.73 \pm 0.30\%$ | $92.92 \pm 0.18\%$ | $86.29 \pm 0.40\%$ |
| ResNet50/CIFAR100 | | | | |
| ProxGen | $73.63 \pm 0.13\%$ | $74.69 \pm 0.58\%$ | $74.03 \pm 0.08\%$ | $67.55 \pm 3.05\%$ |
| RAMDA | $69.90 \pm 1.54\%$ | $69.54 \pm 2.07\%$ | $71.23 \pm 1.39\%$ | $67.48 \pm 1.59\%$ |

Table 3: Weighted group sparsity, validation accuracy and time of ProxSSI and ProxGen for CIFAR10/CIFAR100. We report the average time per epoch using one NVIDIA V100 GPU.

| Algorithm | Accuracy | Sparsity | Time | Accuracy | Sparsity | Time |
|---|---|---|---|---|---|---|
| VGG19/CIFAR10 | | | | VGG19/CIFAR100 | | |
| ProxSSI | $92.8 \pm 0.1\%$ | $88.4 \pm 0.2\%$ | 79s | $67.3 \pm 0.1\%$ | $78.6 \pm 0.3\%$ | 79s |
| ProxGen | $92.8 \pm 0.0\%$ | $86.6 \pm 0.1\%$ | 24s | $68.1 \pm 0.4\%$ | $75.5 \pm 0.2\%$ | 26s |
| ResNet50/CIFAR10 | | | | ResNet50/CIFAR100 | | |
| ProxSSI | $94.0 \pm 0.1\%$ | $83.7 \pm 0.6\%$ | 260s | $73.7 \pm 0.4\%$ | $70.4 \pm 0.7\%$ | 251s |
| ProxGen | $94.1 \pm 0.1\%$ | $80.4 \pm 0.4\%$ | 70s | $73.6 \pm 0.4\%$ | $65.5 \pm 3.6\%$ | 74s |

Huang & Lee (2022); Jelassi & Defazio (2020) such that $\eta_t c_t$ remain a constant, we use a constant momentum value until the final stage, where we gradually increase the momentum value by $c_t = \min(c\sqrt{i}, 1)$, where $i$ counts the training steps executed at this stage. This new momentum strategy is applied to both RAMDA and RMDA in our experiments.

We run each experiment three times with different random initializations and show the mean and standard deviation of the validation predictive performance and the structured sparsity of the final model of all methods.

## 5.1 Subproblem

We start from showing the effectiveness of our proposed subproblem solver for RAMDA and ProxGen. For both approaches, we use Theorem 2 of Deleu & Bengio (2021) to safely screen a portion of groups that will be zero at the optimal subproblem solution, and opt for the PG algorithm discussed in Section 3 to solve the remaining parts. We consider two practical stopping criteria for PG: 1. Running until it reaches the maximum iterations (no early stopping), and 2. Terminate when the subproblem objective is almost non-decreasing (early stopping). For the former, we set the maximum iterations to 100. For the latter, we terminate PG when either $(Q_t(Z^{j-1}) - Q_t(Z^j))/(|Q_t(Z^j)| + 1) < 10^{-8}$ or the iteration cap is reached. Moreover, to ensure incorporation of the preconditioner into ProxGen, we set its minimum PG iterations to 2. We examine how these stopping criteria affect the final model of RAMDA and ProxGen using image classification problems with a smaller scale.

From Table 2, we see that no matter early stopping is used or not, the performance is similar. Given that early stopping is more efficient, we will adopt it in all subsequent experiments.

Next, we compare ProxGen-PG with ProxSSI to examine the efficiency and performance differences between solving the subproblems approximately and exactly in Table 3. These two algorithms are essentially the same except for the subproblem part. We see that our solver is around three times faster than ProxSSI, and the model qualities of the two approaches are similar. In the following experiments, we thus exclude ProxSSI from our comparisons due to its excessively lengthy running time, especially for large-scale models and datasets.

## 5.2 Image Classification

We run a classical computer vision task of training ResNet50 (He et al., 2016) with the ILSVRC 2012 ImageNet dataset (Russakovsky et al., 2015). The result in Table 4 shows that RAMDA attains the best validation accuracy and structured sparsity simultaneously.

We note that some additional experiments in Appendix B show that RAMDA might sometimes perform worse than existing

Table 4: Weighted group sparsity and validation accuracy of different methods for ImageNet.

| Alg. | Accuracy | Group sparsity |
|---|---|---|
| MSGD | $77.14 \pm 0.04\%$ | - |
| ProxSGD | $73.50 \pm 0.20\%$ | $17.54 \pm 1.26\%$ |
| ProxGen | $74.17 \pm 0.08\%$ | $20.29 \pm 0.22\%$ |
| RMDA | $74.47 \pm 0.08\%$ | $25.20 \pm 1.69\%$ |
| RAMDA | $\mathbf{74.53 \pm 0.10\%}$ | $\mathbf{29.19 \pm 0.94\%}$ |

methods on smaller problems like CIFAR10/100. But for such smaller problems, the training cost is not very significant, and one can afford to try more algorithms.

## 5.3 Language Modeling

For language modeling, we train Transformer-XL (base) (Dai et al., 2019) using the WikiText-103 dataset (Merity et al., 2017). Transformer-XL is comprised of embedding and non-embedding layers, and in the PyTorch implementation, the non-embedding layers are built using linear and layer normalization layers. We apply

Table 5: Weighted group sparsity and validation perplexity for training Transformer-XL with WikiText-103.

| Alg. | Perplexity | Group sparsity | Time/epoch |
|---|---|---|---|
| AdamW | $23.00 \pm 0.05$ | - | $6261 \pm 21$s |
| ProxSGD | $27.42 \pm 0.02$ | $33.10 \pm 1.46\%$ | $6167 \pm 12$s |
| ProxGen | $27.49 \pm 0.19$ | $30.47 \pm 0.63\%$ | $6652 \pm 21$s |
| RMDA | $27.10 \pm 0.08$ | $35.99 \pm 2.68\%$ | $6184 \pm 20$s |
| RAMDA | $\mathbf{26.97 \pm 0.10}$ | $\mathbf{36.17 \pm 0.25\%}$ | $6954 \pm 30$s |

group lasso regularization to those linear layers, and present the perplexity and the weighted group sparsity of the models trained by different methods in Table 5. We can see that RAMDA gives the lowest perplexity and the highest structured sparsity.

## 5.4 Speech Synthesis

We consider Tacotron2 (Shen et al., 2018) for speech synthesis on the LJSpeech dataset (Ito & Johnson, 2017). We apply regularization to the convolutional, LSTM, and linear layers of Tacotron2 and show the results in Table 6. Clearly, RAMDA gives the lowest validation loss and the highest group sparsity.

Table 6: Weighted group sparsity and validation loss for training Tacotron2 with the LJSpeech dataset.

| Alg. | Loss | Group sparsity | Time/epoch |
|---|---|---|---|
| AdamW | $0.39 \pm 0.02$ | - | $431 \pm 2$s |
| ProxSGD | $0.50 \pm 0.00$ | $34.29 \pm 1.64\%$ | $431 \pm 0$s |
| ProxGen | $0.45 \pm 0.01$ | $45.63 \pm 0.91\%$ | $438 \pm 2$s |
| RMDA | $0.46 \pm 0.01$ | $45.92 \pm 1.69\%$ | $431 \pm 2$s |
| RAMDA | $\mathbf{0.44 \pm 0.01}$ | $\mathbf{52.85 \pm 1.63\%}$ | $443 \pm 1$s |

## 6 Conclusions

In this work, we propose a regularized dual averaging method with adaptiveness for training structured neural networks. Our method is particularly useful for language models and speech recognition and proven to outperform state of the art on modern architectures including LSTM and transformers as well as the ImageNet problem. We also propose a subroutine to approximately solve the regularized subproblem for both our method and an existing framework with strong convergence guarantees. Extensive experiments on group sparsity show that our subproblem solver can greatly reduce the training time for existing methods, and our proposed algorithm RAMDA achieves simultaneously higher structured sparsity ratio and better prediction performance than existing methods. Our implementation will be released for public use.

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

# Appendices

## Table of Contents

## A  PROOFS

This section provides proofs of the theoretical results stated in Section 4. We restate these results and provide their corresponding proofs right after each statement.

**Theorem 1.** *Assume that* (4) *and* (6) *has at least one optimal solution, with the optimal function value being finite. Given any $\epsilon_t > 0$, the number of iterations of Algorithm* 2 *needed for satisfying* (7) *for both* (4) *and* (6) *is $O(\epsilon_t^{-1})$ when $\psi$ is convex and $O(\epsilon_t^{-2})$ when $\psi$ is nonconvex.*

*Proof.* We use the notation

$$\bar{Q}_t(Z) = f_t(Z) + \psi(Z)$$

to unify the two objective function $Q_t/\alpha_t$ and $\hat{Q}_t$, where $f_t$ is the smooth part that has $L$-Lipschitz continuous gradients.

At each iteration of PG, it solves the following subproblem

$$Z^{j+1} \in \arg\min_{Z} \langle \nabla f_t(Z^j), Z - Z^j \rangle + \frac{1}{2\theta_t} \| Z - Z^j \|^2 + \psi(Z),$$

and thus from the first-order optimality conditions, clearly it satisfies

$$\nabla f_t(Z^{j+1}) - \nabla f_t(Z^j) - \frac{1}{\theta_t} \left( Z^{j+1} - Z^j \right) \in \partial \bar{Q}_t(Z^{j+1}).$$

We thus have from the Lipschitz continuity of $\nabla f_t$ that

$$\min_{s \in \partial \bar{Q}_t(Z^{j+1})} \| s \| \leq \left\| \nabla f_t(Z^{j+1}) - \nabla f_t(Z^j) - \frac{1}{\theta_t} \left( Z^{j+1} - Z^j \right) \right\| \leq \left( L + \theta_t^{-1} \right) \| Z^{j+1} - Z^j \|. \tag{9}$$

Note that $\bar{Q}_t$ is lower bounded, say by $\bar{Q}_t^*$, and has at least one solution $Z^*$ (unique when $\psi$ is convex).

In the case that $\psi$ is convex, we know that $\theta_t = 1/L$, and standard analysis of proximal gradient (see, for example, Beck, 2017, Theorem 10.27) gives that

$$L \| Z^{j+1} - Z^j \| \leq \frac{2L \| Z^0 - Z^* \|}{j}, \forall j \geq 0. \tag{10}$$

Therefore, the combination of (9) and (10) shows that it takes $O(\epsilon_t^{-1})$ iterations for PG to reach the required precision of $\epsilon_t$.

When $\psi$ is nonconvex, we have that $\theta_t = 1/(2L)$ and standard analysis (Beck, 2017, Theorem 10.15) gives

$$\min_{k=0,1,\dots,j} \left\| Z^{j+1} - Z^j \right\| \leq \frac{C}{\sqrt{j}} \tag{11}$$

for some constant $C$ depending on $L$ and $\bar{Q}_t(W^t) - \bar{Q}_t^*$. Therefore, (11) and (9) show that it takes $O(\epsilon_t^{-2})$ iterations to reach the desired precision. $\qquad \square$

**Theorem 2.** *Consider $\{\tilde{W}^t\}$ generated by Algorithm 1 with (7) for (1), with $\{c_t\}$ and $\{\epsilon_t\}$ satisfying $\sum c_t = \infty$ and (8). Assume for any $\xi \sim \mathcal{D}$, $f_\xi$ is L-Lipschitz-continuously-differentiable almost surely for some $L$, so the expectation is also L-Lipschitz-continuously-differentiable, there is $C \geq 0$ such that $\mathbb{E}_{\xi_t \sim \mathcal{D}} \left\| \nabla f_{\xi_t}\left(W^{t-1}\right) \right\|^4 \leq C$ for all $t$, and that the set of stationary points $\mathcal{Z} := \{W \mid 0 \in \partial F(W)\}$ is nonempty. For any given $W^0$, consider the event that $\{\tilde{W}^t\}$ converges to a point $W^*$ (each event corresponds to a different $W^*$). If $\partial \psi$ is outer semicontinuous at $W^*$, this event has a nonzero probability, and $\{\eta_t\}$ satisfy*

$$\sum s_t \alpha_t^{-1} = \infty, \quad \sum \left(s_t \alpha_t^{-1}\right)^2 < \infty, \quad \left\| W^{t+1} - W^t \right\| \left(s_t \alpha_t^{-1}\right)^{-1} \xrightarrow{a.s.} 0, \tag{12}$$

*then we have that $W^* \in \mathcal{Z}$ with probability one conditional on this event. Moreover, $\{W^t\}$ also converges to this stationary point $W^*$.*

*Proof.* First, we prove that when $\{\tilde{W}^t\}$ converges to $W^*$, $W^t$ also converges to $W^*$. Indeed, from (5), we have that

$$\left\| W^t - W^* \right\| \leq (1 - c_t)\left\| W^{t-1} - W^* \right\| + c_t \left\| \tilde{W}^t - W^* \right\|, \tag{13}$$

Assume that it is not true, then there is $\delta > 0$ such that $\|W^t - W^*\| \geq \delta$ for all $t$. Consider some $t_0$ such that $\left\| \tilde{W}^t - W^* \right\| \leq \delta/2$ for all $t \geq t_0$. (13) then gives

$$\left\| W^{t-1} - W^* \right\| - \left\| W^t - W^* \right\| \geq c_t \left\| W^{t-1} - W^* \right\| - \frac{c_t \delta}{2} \geq \frac{c_t \delta}{2}, \quad \forall t > t_0. \tag{14}$$

By summing (14) from $t = t_0 + 1$ to $t = T$ for any $T > t_0 + 1$, we obtain

$$\left\| W^{t_0} - W^* \right\| \geq \frac{\delta \sum_{t=t_0+1}^{T} c_t}{2}.$$

However, since $\sum c_t = \infty$, the right-hand side approaches infinity as $T$ goes to infinity, which cannot hold true because $\|W^{t_0} - W^*\|$ is a finite value that does not change with $T$. This part is thus proven true by contradiction.

Next, consider the update of $\alpha_t^{-1} U^t$, we can see from (2) that

$$\frac{U^t}{\alpha_t} = \frac{\alpha_{t-1}}{\alpha_t} \frac{U^{t-1}}{\alpha_{t-1}} + \frac{s_t \nabla f_{\xi_t}(W^{t-1})}{\alpha_t} = \left(1 - \frac{s_t}{\alpha_t}\right) \frac{U^{t-1}}{\alpha_{t-1}} + \frac{s_t}{\alpha_t} \nabla f_{\xi_t}(W^{t-1}). \tag{15}$$

Moreover, the assumptions on $\eta_t$ satisfies all the required conditions of Lemma 1 of Ruszczyński (1980). We therefore apply Lemma 1 of Ruszczyński (1980) to conclude that

$$\frac{U^t}{\alpha_t} \xrightarrow{a.s.} \nabla \mathbb{E}_{\xi \sim \mathcal{D}} \left[ f_\xi\left(W^t\right) \right] \circ \nabla \mathbb{E}_{\xi \sim \mathcal{D}} \left[ f_\xi\left(W^t\right) \right]. \tag{16}$$

The update for $\alpha_t^{-1} V^t$ has a form analogous to (15), and we have from Jensen's inequality that

$$\mathbb{E}_{\xi_t \sim \mathcal{D}} \left\| \nabla f_{\xi_t}\left(W^{t-1}\right) \right\|^2 \leq \sqrt{\mathbb{E}_{\xi_t \sim \mathcal{D}} \| \nabla f_{\xi_t}(W^{t-1}) \|^4} \leq \sqrt{C},$$

implying that the second moment is also bounded in expectation. We can therefore also apply Lemma 1 of Ruszczyński (1980) to $\alpha_t^{-1} V^t$ and conclude that

$$\frac{V^t}{\alpha_t} \xrightarrow{a.s.} \nabla \mathbb{E}_{\xi \sim \mathcal{D}} \left[ f_\xi\left(W^t\right) \right]. \tag{17}$$

We further notice that the union of two almost surely events is still an almost surely event.

From (4) and (7), we can see that there is a sequence $\{z_t\}$ such that

$$-\left(\frac{V^t}{\alpha_t} + \frac{z_t}{\alpha_t} + \frac{P^t}{\alpha_t}(\tilde{W}^t - W^0)\right) \in \partial\psi(\tilde{W}^t), \quad \|z_t\| \le \epsilon_t. \tag{18}$$

Our assumption in (12) implies that $\alpha_t \to \infty$, which together with (8) shows that

$$\frac{z_t}{\alpha_t} \to 0. \tag{19}$$

From (17), that $\nabla\mathbb{E}_{\xi\sim\mathcal{D}}\left[f_\xi(W)\right]$ is Lipschitz continuous, and that $W^t \to W^*$ (which we have proven in the first part), we see that

$$\frac{V^t}{\alpha_t} \xrightarrow{\text{a.s.}} \nabla\mathbb{E}_{\xi\sim\mathcal{D}}\left[f_\xi(W^*)\right]. \tag{20}$$

For the third term, we have from (3) and (16) that

$$\frac{P^t}{\alpha_t} = \alpha_t^{-\frac{2}{3}} \text{Diag}\left(\sqrt[3]{\frac{U^t}{\alpha_t}}\right) + \frac{\epsilon}{\alpha_t}I.$$

Again since $\alpha_t \to \infty$, the second term of the equation above converges to 0. Therefore, by (16), we obtain that

$$\frac{P^t}{\alpha_t} \xrightarrow{\text{a.s.}} \alpha_t^{-\frac{2}{3}} \text{Diag}\left(\sqrt[3]{\nabla\mathbb{E}_{\xi\sim\mathcal{D}}\left[f_\xi(W^t)\right] \circ \nabla\mathbb{E}_{\xi\sim\mathcal{D}}\left[f_\xi(W^t)\right]}\right).$$

Again from the continuity of $\nabla\mathbb{E}_{\xi\sim\mathcal{D}}\left[f_\xi(W)\right]$ and that $\alpha_t \to \infty$, we conclude that

$$\frac{P^t}{\alpha_t} \xrightarrow{\text{a.s.}} \alpha_t^{-\frac{2}{3}} \text{Diag}\left(\sqrt[3]{\nabla\mathbb{E}_{\xi\sim\mathcal{D}}\left[f_\xi(W^*)\right] \circ \nabla\mathbb{E}_{\xi\sim\mathcal{D}}\left[f_\xi(W^*)\right]}\right) \xrightarrow{\text{a.s.}} 0. \tag{21}$$

Finally, using the outer semicontinuity of $\partial\psi(W)$ at $W^*$, we conclude from (18)–(21) that

$$0 \in \nabla\mathbb{E}_{\xi\sim\mathcal{D}}\left[f_\xi(W^*)\right] + \lim_{t\to\infty}\psi(\tilde{W}^t) \subseteq \nabla\mathbb{E}_{\xi\sim\mathcal{D}}\left[f_\xi(W^*)\right] + \psi(W^*) = \partial F(W^*)$$

with probability one, showing that $W^*$ is a stationary point almost surely.

$\square$

**Theorem 3.** *Consider Algorithm 1 with the conditions in Theorem 2 satisfied. Consider the event of $\{\tilde{W}^t\}$ converging to a certain point $W^*$ as in Theorem 2, if the probability of this event is nonzero; $\psi$ is prox-regular and subdifferentially continuous at $W^*$ and partly smooth at $W^*$ relative to the active $\mathcal{C}^2$ manifold $\mathcal{M}_{W^*}$; $\partial\psi$ is outer semicontinuous at $W^*$; and the nondegeneracy condition*

$$-\nabla f(W^*) \in \text{relint } \partial\psi(W^*) \tag{22}$$

*holds at $W^*$, then conditional on this event, almost surely there is $T_0 \ge 0$ such that*

$$\tilde{W}^t \in \mathcal{M}_{W^*}, \quad \forall t \ge T_0. \tag{23}$$

*In other words, the active manifold at $W^*$ is identified by the iterates of Algorithm 1 after a finite number of iterations almost surely.*

*Proof.* From (18), there exists a sequence $\{Y^t\}$ such that

$$Y^t \in \partial\psi(\tilde{W}^t), \quad \frac{V^t}{\alpha_t} + \frac{z_t}{\alpha_t} + \frac{P^t}{\alpha_t}(\tilde{W}^t - W^0) + Y^t = 0, \quad \forall t. \tag{24}$$

For notational ease, we denote

$$f(W) := \mathbb{E}_{\xi\sim\mathcal{D}}\left[f_\xi(W)\right]. \tag{25}$$

We thus have

$$\nabla f(\tilde{W}^t) + Y^t \in \partial F(\tilde{W}^t), \quad \forall t.$$

From (24), we then get

$$\nabla f(\tilde{W}^t) - \frac{V^t}{\alpha_t} - \frac{z_t}{\alpha_t} - \frac{P^t}{\alpha_t}(\tilde{W}^t - W^0) \in \partial F(\tilde{W}^t). \tag{26}$$

We aim to show that

$$\operatorname{dist}(0, \partial F(\tilde{W}^t)) \coloneqq \min_{Y \in \partial F(\tilde{W}^t)} \|Y\|$$

converges to 0 almost surely. From (26), we have

$$
\begin{aligned}
\operatorname{dist}(0, \partial F(\tilde{W}^t)) &\leq \left\| \nabla f(\tilde{W}^t) - \frac{V^t}{\alpha_t} - \frac{z_t}{\alpha_t} - \frac{P^t}{\alpha_t}(\tilde{W}^t - W^0) \right\| \\
&\leq \left\| \nabla f(\tilde{W}^t) - \frac{V^t}{\alpha_t} \right\| + \left\| \frac{z_t}{\alpha_t} \right\| + \left\| \frac{P^t}{\alpha_t}(\tilde{W}^t - W^0) \right\| \\
&\leq \left\| \nabla f(\tilde{W}^t) - \frac{V^t}{\alpha_t} \right\| + \frac{\epsilon_t}{\alpha_t} + \left\| \frac{P^t}{\alpha_t}(\tilde{W}^t - W^0) \right\|. 
\end{aligned} \tag{27}
$$

From (16) and (17), there are $\{A_t\}$ and $\{B_t\}$ such that

$$
\begin{cases}
\frac{V^t}{\alpha_t} &= \nabla f(W^t) + A_t, \quad \|A_t\| \xrightarrow{\text{a.s.}} 0 \\
\frac{P^t}{\alpha_t} &= \alpha_t^{-\frac{2}{3}} \operatorname{Diag}\left( \sqrt[3]{\nabla f(W^t) \circ \nabla f(W^t)} \right) + B_t, \quad \|B_t\| \xrightarrow{\text{a.s.}} 0.
\end{cases} \tag{28}
$$

Substituting the above two equations back to (27), we obtain

$$
\begin{aligned}
&\operatorname{dist}(0, \partial F(\tilde{W}^t)) \\
&\leq \left\| \nabla f(\tilde{W}^t) - \nabla f(W^t) \right\| + \|A_t\| + \frac{\epsilon_t}{\alpha_t} + \left( \alpha_t^{-\frac{2}{3}} \left\| \sqrt[3]{\nabla f(W^t) \circ \nabla f(W^t)} \right\| + \|B_t\| \right) \left\| \tilde{W}^t - W^0 \right\| \\
&\leq L \left\| \tilde{W}^t - W^t \right\| + \|A_t\| + \frac{\epsilon_t}{\alpha_t} + \left( \alpha_t^{-\frac{2}{3}} \left\| \sqrt[3]{\nabla f(W^t) \circ \nabla f(W^t)} \right\| + \|B_t\| \right) \left\| \tilde{W}^t - W^0 \right\|. \tag{29}
\end{aligned}
$$

From Theorem 2, we know that $\tilde{W}^t$ and $W^t$ both converge to $W^*$, so

$$\left\| \tilde{W}^t - W^t \right\| \leq \left\| \tilde{W}^t - W^* \right\| + \left\| W^t - W^* \right\| \to 0.$$

From (8) and (12), we know that $\epsilon_t/\alpha_t \to 0$. Because $\tilde{W}^t \to W^*$, we have that

$$\left\| \tilde{W}^t - W^0 \right\| \to \left\| W^* - W^0 \right\| < \infty.$$

From $W^t \to W^*$, we have that

$$\left\| \sqrt[3]{\nabla f(W^t) \circ \nabla f(W^t)} \right\| \to \left\| \sqrt[3]{\nabla f(W^*) \circ \nabla f(W^*)} \right\| < \infty.$$

Combining these results with (29), we conclude that

$$L \left\| \tilde{W}^t - W^t \right\| + \|A_t\| + \frac{\epsilon_t}{\alpha_t} + \left( \alpha_t^{-\frac{2}{3}} \left\| \sqrt[3]{\nabla f(W^t) \circ \nabla f(W^t)} \right\| + \|B_t\| \right) \left\| \tilde{W}^t - W^0 \right\| \xrightarrow{\text{a.s.}} 0,$$

proving that

$$\operatorname{dist}(0, \partial F(\tilde{W}^t)) \xrightarrow{\text{a.s.}} 0.$$

On the other hand, since $f$ is continuous and $\psi$ is subdifferentially continuous at $W^*$, $\tilde{W}^t \to W^*$, and that $\nabla f(\tilde{W}^t) + Y_t \xrightarrow{\text{a.s.}} 0 \in \partial F(W^*)$, we know that $F(\tilde{W}^t) \xrightarrow{\text{a.s.}} F(W^*)$ as well. Therefore, we can apply (Lemma 1 Lee, 2023) to prove that (23) indeed holds for some $T_0 < \infty$. $\qquad \square$

## A.1 Convergence Result for ProxGen

We next discuss the convergence result for the framework of Yun et al. (2021) with inexactness being added. For consistency, we first use our notations to introduce their framework with our inexactness condition added in Algorithm 3.

In their analysis, Yun et al. (2021) made the following four assumptions, and we will follow these assumptions using the notation (25).

**Algorithm 3:** ProxGen $(W^0, T, T_2, \{\eta_t\}, \{\rho_t\}, \{c_t\}, \{\epsilon_t\}, \{b_t\}, \delta I)$

---

$m_0 \leftarrow 0$
**for** $t = 1, \ldots, T$ **do**
    Sample $\xi_t \sim \mathcal{D}$ with batch size $b_t$
    $G^t \leftarrow \nabla f_{\xi_t}(W^{t-1})$
    $m_t \leftarrow \rho_t m_{t-1} + (1 - \rho_t) G^t$
    Construct $P^t$ satisfying $P^t \succeq \delta I$
    $\theta_t \leftarrow 1/\|P^t\|_2$
    Compute $W^{t+1}$ by roughly solving (6) that satisfies (7) with $Q_t$ replaced by $\hat{Q}_t$ and
    $\tilde{W}^t$ replaced by $W^{t+1}$, using $\text{PG}(W^t, W^t, m_t, \eta_t^{-1} P^t, \theta_t, T_2, \epsilon_t)$
**output:** $W^T$

---

**(C-1)** The loss function $f$ is $L$-Lipschitz-continuously-differentiable and lower-bounded.

**(C-2)** The stochastic gradient $G^t = \nabla f_{\xi_t}(W^{t-1})$ is unbiased, and has a bounded variance.

$$E_{\xi_t \sim \mathcal{D}}[G^t]] = \nabla f(W^{t-1}), E_{\xi_t \sim D}\left[\left\|G^t - \nabla f(W^{t-1})\right\|^2\right] \leq \frac{\sigma^2}{b_t},$$

    where $b_t$ is the batch size of $\xi_t$.

**(C-3)** $\left\|W^{t+1} - W^t\right\| \leq D, \|G^t\| \leq G, \rho_t = \rho_0 \mu^{t-1}$ for all $t$, where $\rho_0, \mu \in [0, 1)$ and $D, G > 0$.

**(C-4)** $\left\|\eta_t^{-1} P^t\right\|_2 \leq 1/\gamma < \infty$.

We also have from our reformulation that there is $\delta > 0$ such that

$$P^t \geq \delta, \eta_t \leq \frac{\delta}{3L}. \tag{30}$$

**Theorem 4.** *For the framework in Yun et al. (2021) with the subproblem solved approximately by Algorithm 2 and that (7) holds with $\{\epsilon_t\}$ satisfying (8). Then Theorem 1 of Yun et al. (2021) still holds, but with the constants $\{Q_i\}$ being also dependent on $\sum_{t=0}^{\infty} \epsilon_t^2$.*

*Proof.* The major flow of our proof follows that of Yun et al. (2021) but with suitable modifications to accommodate the inexactness condition in the subproblem solving. It is clear from (Yun et al., 2021, Lemma 1) that $\|m_t\| \leq G$ for all $t$.

By (6) and (7), we have

$$0 \in z_t + (1 - \rho_t)g_t + \rho_t m_{t-1} + \partial \psi(W^t) + \frac{1}{\eta_t}(P^t)(W^t - W^{t-1}), \quad \|z_t\| \leq \epsilon_t,$$

leading to

$$\nabla f(W^t) - z_t - (1 - \rho_t)g_t - \rho_t m_{t-1} - \frac{1}{\eta_t}(P^t)(W^t - W^{t-1}) \in \partial F(W^t). \tag{31}$$

We thus have from (31) and (C-4) that

$\text{dist}(0, \partial F(W^t))^2$

$\leq \left\|z_t + (1 - \rho_t)g_t - \nabla f(W^t) + \rho_t m_{t-1} + (W^t - W^{t-1}) + \frac{1}{\eta_t}(P^t)(W^t - W^{t-1}) - (W^t - W^{t-1})\right\|^2$

$\leq 4\left\|(1 - \rho_t)g_t - \nabla f(W^t) + \rho_t m_{t-1} + (W^t - W^{t-1})\right\|^2 + 4\epsilon_t^2 + 4\left\|\frac{1}{\eta_t}(P^t)(W^t - W^{t-1})\right\|^2$

$\quad + 4\left\|(W^t - W^{t-1})\right\|^2$

$\leq 4\underbrace{\left\|(1 - \rho_t)g_t - \nabla f(W^t) + \rho_t m_{t-1} + (W^t - W^{t-1})\right\|^2}_{T_1} + 4\left(\frac{1}{\gamma^2} + 1\right)\|W^t - W^{t-1}\|^2 + 4\epsilon_t^2.$

$$\tag{32}$$

We will separately bound the quantities $T_1$ and $\left\|W^t - W^{t-1}\right\|^2$ below.

From the subproblem objective requirement in (7), we also get

$$\left\langle (1-\rho_t)g_t + \rho_t m_{t-1}, W^t - W^{t-1}\right\rangle + \psi(W^t) + \frac{1}{2\eta_t}\langle W^t - W^{t-1}, P^t(W^t - W^{t-1})\rangle \tag{33}$$
$$\leq \psi(W^{t-1}).$$

From (C-1), we have

$$f(W^t) \leq f(W^{t-1}) + \langle \nabla f(W^{t-1}), W^t - W^{t-1}\rangle + \frac{L}{2}\|W^t - W^{t-1}\|^2. \tag{34}$$

Summing (33) and (34) gives

$$\left\langle (1-\rho_t)g_t - \nabla f(W^{t-1}) + \rho_t m_{t-1}, W^t - W^{t-1}\right\rangle + \left\|W^t - W^{t-1}\right\|^2_{\frac{P^t}{2\eta_t} - \frac{L}{2}I} \tag{35}$$
$$\leq F(W^{t-1}) - F(W^t).$$

Note that $(2\eta_t)^{-1}P^t - LI/2 \succeq 0$ from (30) so the second term in (35) is nonnegative. (35) together with (C-3) then leads to

$$\|W^t - W^{t-1}\|^2_{\frac{P^t}{2\eta_t} - \frac{L}{2}I}$$
$$\leq F(W^{t-1}) - F(W^t) - \left\langle g_t - \nabla f(W^{t-1}), W^t - W^{t-1}\right\rangle + \langle \rho_t g_t, W^t - W^{t-1}\rangle - \langle \rho_t m_{t-1}, W^t - W^{t-1}\rangle$$
$$\leq F(W^{t-1}) - F(W^t) + \frac{1}{2L}\|g_t - \nabla f(W^{t-1})\|^2 + \frac{L}{2}\|W^t - W^{t-1}\|^2 + \frac{\rho_t^2}{2L}\|g_t\|^2 + \frac{L}{2}\|W^t - W^{t-1}\|^2$$
$$\quad + \|\rho_t m_{t-1}\|\|W^t - W^{t-1}\|$$
$$\leq F(W^{t-1}) - F(W^t) + \rho_0\mu^{t-1}DG + \frac{\rho_0^2\mu^{2(t-1)}G^2}{2L} + L\|W^t - W^{t-1}\|^2 + \frac{1}{2L}\|g_t - \nabla f(W^{t-1})\|^2.$$

Summing it over $t = 1, 2, \ldots, T$ and utilizing the assumption that the step sizes are non-increasing then give

$$\left(\frac{\delta}{2\eta_0} - \frac{3}{2}L\right)\sum_{t=1}^{T}\|W^t - W^{t-1}\|^2 \leq \Delta + C_1 + \frac{1}{2L}\sum_{t=1}^{T}\|g_t - \nabla f(W^{t-1})\|^2,$$

where

$$\Delta := F(W^0) - \min_W F(W), \quad C_1 := \frac{\rho_0 DG}{1-\mu} + \frac{\rho_0^2 G^2}{2L(1-\mu^2)}.$$

From the inequality above, we obtain

$$\sum_{t=1}^{T}\|W^t - W^{t-1}\|^2 \leq H_1 + H_2\sum_{t=1}^{T}\|g_t - \nabla f(W^{t-1})\|^2 \tag{36}$$

for some constants $H_1, H_2$ depending on $L, \Delta, \delta, \eta_0$, and $C_1$. From (35), we have

$$\left\langle (1-\rho_t)g_t - \nabla f(W^t) + \rho_t m_{t-1}, W^t - W^{t-1}\right\rangle$$
$$\leq F(W^{t-1}) - F(W^t) - \left\langle \nabla f(W^t) - \nabla f(W^{t-1}), W^t - W^{t-1}\right\rangle - \left\|W^t - W^{t-1}\right\|^2_{\frac{1}{2\eta_t}(P^t) - \frac{L}{2}I}$$
$$\leq F(W^{t-1}) - F(W^t) - \left\langle \nabla f(W^t) - \nabla f(W^{t-1}), W^t - W^{t-1}\right\rangle.$$

Therefore, we obtain

$$T_1 = \|(1-\rho_t)g_t - \nabla f(W^t) + \rho_t m_{t-1}\|^2 + \|W^t - W^{t-1}\|^2$$
$$\quad + 2\left\langle (1-\rho_t)g_t - \nabla f(W^t) + \rho_t m_{t-1}, W^t - W^{t-1}\right\rangle$$
$$\leq \|(1-\rho_t)g_t - \nabla f(W^{t-1}) + \nabla f(W^{t-1}) - \nabla f(W^t) + \rho_t m_{t-1}\|^2 + \|W^t - W^{t-1}\|^2$$
$$\quad + F(W^{t-1}) - F(W^t) - \left\langle \nabla f(W^t) - \nabla f(W^{t-1}), W^t - W^{t-1}\right\rangle$$
$$\leq 4\|g_t - \nabla f(W^{t-1})\|^2 + 4L^2\|W^t - W^{t-1}\|^2 + 4\rho_t^2(\|m_{t-1}\|^2 + \|g_t\|^2) + \|W^t - W^{t-1}\|^2$$
$$\quad + F(W^{t-1}) - F(W^t) + L\|W^t - W^{t-1}\|^2$$
$$\leq F(W^{t-1}) - F(W^t) + 8\rho_0^2\mu^{2(t-1)}G^2 + \left(1 + L + 4L^2\right)\|W^t - W^{t-1}\|^2 + 4\|g_t - \nabla f(W^{t-1})\|^2. \tag{37}$$

Let $C_2 := 2 + L + 4L^2 + \gamma^{-2}$ and insert (37) into (32), we get

$$\text{dist}(0, \partial F(W^t))^2$$
$$\leq 4\left( F(W^{t-1}) - F(W^t) + 8\rho_0^2 \mu^{2(t-1)} G^2 + C_2 \|W^t - W^{t-1}\|^2 + 4\|g_t - \nabla f(W^{t-1})\|^2 + \epsilon_t^2 \right). \tag{38}$$

Therefore, by letting $S := \sum_{t=1}^{\infty} \epsilon_t^2$ and noting that $S < \infty$, we have from (38) and (C-2) that

$$\mathbb{E}_{a, \xi_1, \ldots, \xi_T}[\text{dist}(0, \partial F(W^a))^2]$$
$$\leq \frac{1}{T} \sum_{t=1}^{T} \mathbb{E}\left[ \left\| (1 - \rho_t) g_t - \nabla f(W^t) + z_t + \rho_t m_{t-1} + \frac{1}{\eta_t}(P^t)(W^t - W^{t-1}) \right\|^2 \right]$$
$$\leq \frac{4}{T}\left( \Delta + \frac{8\rho_0^2 G^2}{1 - \mu^2} + 4 \sum_{t=1}^{T} \mathbb{E}\|g_t - \nabla f(W^{t-1})\|^2 + C_2 \sum_{t=1}^{T} \mathbb{E}\|W^t - W^{t-1}\|^2 + \sum_{t=1}^{T} \epsilon_t^2 \right)$$
$$\leq \frac{4}{T}\left( \Delta + \frac{8\rho_0^2 G^2}{1 - \mu^2} + 4\sigma^2 \sum_{t=1}^{T} \frac{1}{b_t} + C_2(H_1 + H_2\sigma^2 \sum_{t=1}^{T} \frac{1}{b_t}) + S \right)$$
$$\leq \frac{Q_1}{T} \sum_{t=1}^{T} \frac{1}{b_t} + \frac{Q_2 \Delta}{T} + \frac{Q_3}{T},$$

for some constants $Q_1, Q_2, Q_3$ dependent on $\{\eta_0, \delta, \Delta, L, D, G, \rho_0, \mu, \gamma, S\}$, but not on $T$. This proves our theorem. $\square$

## B  ADDITIONAL EXPERIMENTS FOR COMPUTER VISION

In this section, we compare RAMDA with other methods on image classification with smaller datasets. They are:

1. Logistic regression (neural network with no hidden layer) with the MNIST dataset (LeCun et al., 1998).

2. A modified VGG19 (Simonyan & Zisserman, 2015) with the CIFAR10 dataset (Krizhevsky, 2009).

3. The same VGG19 with the CIFAR100 dataset (Krizhevsky, 2009).

4. A modified ResNet50 (He et al., 2016) with the CIFAR10 dataset.

5. The same ResNet50 with the CIFAR100 dataset.

The results are shown in Table 7. In the logistic regression task, we only perform a single run as it is a convex problem. Moreover, when dealing with ProxSSI, ProxGen, and ProxSGD in the logistic regression problem, whose sparsity levels are highly unstable over iterations, we report their highest weighted group sparsity over all epochs, but for all other problems, we report the group sparsity level of the final output.

## C  PLOTS OF SPARSITY LEVEL AND VALIDATION ACCURACY OVER EPOCHS

In this section, we provide the plots of predictive ability and structured sparsity over epochs for all conducted experiments in Fig. 1. In the plot for Transformer-XL, one step processes ten batches, and for our batch size of 64, one epoch consists of 8,401 batches. These experiments are:

1. ResNet50 with the ILSVRC 2012 ImageNet dataset.

Table 7: Group sparsity and validation accuracy of different methods on image classification with smaller datasets.

| Algorithm | Validation accuracy | Group sparsity |
|---|---|---|
| Logistic Regression/MNIST | | |
| ProxSGD | 91.31% | 39.29% |
| ProxSSI | 91.31% | 39.92% |
| ProxGen | 91.31% | 39.92% |
| RMDA | 91.34% | 57.02% |
| RAMDA | **91.35%** | **57.40%** |
| VGG19/CIFAR10 | | |
| MSGD | $93.95 \pm 0.14\%$ | - |
| ProxSGD | $92.82 \pm 0.09\%$ | $82.76 \pm 5.42\%$ |
| ProxSSI | $92.81 \pm 0.15\%$ | $88.40 \pm 0.23\%$ |
| ProxGen | $92.83 \pm 0.05\%$ | $86.64 \pm 0.12\%$ |
| RMDA | $\mathbf{93.13 \pm 0.10\%}$ | $\mathbf{90.22 \pm 0.06\%}$ |
| RAMDA | $92.89 \pm 0.13\%$ | $86.31 \pm 0.31\%$ |
| VGG19/CIFAR100 | | |
| MSGD | $74.07 \pm 0.05\%$ | - |
| ProxSGD | $\mathbf{71.96 \pm 0.15\%}$ | $72.34 \pm 11.9\%$ |
| ProxSSI | $67.29 \pm 0.06\%$ | $78.58 \pm 0.34\%$ |
| ProxGen | $68.13 \pm 0.36\%$ | $75.46 \pm 0.17\%$ |
| RMDA | $\mathbf{71.96 \pm 0.31\%}$ | $\mathbf{80.88 \pm 0.11\%}$ |
| RAMDA | $70.47 \pm 0.25\%$ | $65.19 \pm 0.77\%$ |
| ResNet50/CIFAR10 | | |
| MSGD | $95.54 \pm 0.19\%$ | - |
| ProxSGD | $92.36 \pm 0.05\%$ | $82.18 \pm 2.67\%$ |
| ProxSSI | $94.04 \pm 0.12\%$ | $83.67 \pm 0.63\%$ |
| ProxGen | $94.07 \pm 0.12\%$ | $80.45 \pm 0.45\%$ |
| RMDA | $\mathbf{95.11 \pm 0.11\%}$ | $\mathbf{85.64 \pm 0.12\%}$ |
| RAMDA | $93.85 \pm 0.10\%$ | $81.99 \pm 1.26\%$ |
| ResNet50/CIFAR100 | | |
| MSGD | $79.49 \pm 0.49\%$ | - |
| ProxSGD | $74.54 \pm 0.58\%$ | $49.29 \pm 5.91\%$ |
| ProxSSI | $73.65 \pm 0.39\%$ | $70.38 \pm 0.74\%$ |
| ProxGen | $73.63 \pm 0.43\%$ | $65.51 \pm 3.58\%$ |
| RMDA | $\mathbf{75.62 \pm 0.19\%}$ | $\mathbf{79.97 \pm 0.27\%}$ |
| RAMDA | $69.23 \pm 0.86\%$ | $68.65 \pm 1.83\%$ |

2. Transformer-XL with the WikiText-103 dataset.

3. Tacotron2 with the LJSpeech dataset.

4. Logistic Regression with the MNIST dataset.

5. A modified VGG19 with the CIFAR10 dataset.

6. The same VGG19 with the CIFAR100 dataset.

7. A modified ResNet50 with the CIFAR10 dataset.

8. The same ResNet50 with the CIFAR100 dataset.

We further see the zoomed-in sparsity plots in Fig. 2 of the stable sparsity level of RAMDA. These plots corroborates our theory that RAMDA is indeed capable of manifold identification, while achieving competitive prediction performance. On the other hand, in the absence of manifold identification guarantees, the sparsity levels of ProxSGD, ProxSSI and ProxGen exhibit oscillations that are sometimes drastic. We note that for the largest problems Tacotron2 and Transformer-XL, the sparsity levels of RAMDA were still gradually

increasing even at the final epochs. This suggests that if we are willing to run the algorithm for longer, it is possible that the structured sparsity can be further improved.

## D  EXPERIMENT WITH NUCLEAR-NORM REGULARIZATION

In this appendix, we conduct some preliminary experiments with a different regularizer to showcase that the proposed RAMDA can be applied to structures beyond sparsity. We consider the structure of each layer being low-rank, induced by imposing a nuclear-norm regularizer on each layer individually by treating each layer as a matrix. Given a matrix $X \in \mathbb{R}^{m \times n}$ of rank $r \leq \min\{m, n\}$ with its singular value decomposition (SVD) $X = U\Sigma V^\top$, where $U \in \mathbb{R}^{m \times r}$, $V \in \mathbb{R}^{n \times r}$ are orthogonal and the positive definite diagonal matrix $\Sigma \in \mathbb{R}^{r \times r}$ represents the nonzero singular values of $X$, the nuclear norm of $X$ is computed by

$$\|X\|_* = \sum_{i=1}^{r} \Sigma_{i,i},$$

and the corresponding proximal operator for $\lambda > 0$ is

$$\mathrm{prox}_{\lambda \|\cdot\|_*}(X) = U\hat{\Sigma}V^\top, \text{ where } \hat{\Sigma}_{i,i} = \max\{0, \Sigma_{i,i} - \lambda\}.$$

Given a point $X^*$ with rank $r^*$, the active manifold of the nuclear norm at $X^*$ is

$$\mathcal{M}(X^*) = \{Y \mid \mathrm{rank}(Y) = r^*\}.$$

Using low-rankness to condense neural networks is itself an interesting research topic, but conducting full SVDs could be rather time-consuming, so applying this structure to larger problems is a challenging but potentially useful one. How to exploit this structure for prediction acceleration and to make the training more efficient, possibly using iterative methods to compute approximate SVDs, is an interesting topic we plan to investigate in the near future, and here we conduct a preliminary experiment for showing that our method is also applicable to other structures.

We first consider training a simple neural network with six fully-connected layers using the FashionMNIST dataset (Xiao et al., 2017). Since this is a rather small-scale problem and this is a image classification problem, we do not expect RAMDA to outperform non-adaptive methods, especially the RMDA method that is also able to identify the active manifold. The purpose of this experiment is to demonstrate the possibilities of structures beyond sparsity. The results are shown in Table 8. As we have anticipated, RAMDA is indeed slightly worse than RMDA regarding the low-rank level and the prediction accuracy, but it is still competitive and outperforms ProxGen and ProxSGD. This exemplifies the potential of RAMDA as well as RMDA for training neural networks with other useful structures.

We also conduct an experiment on pretraining a modified vision transformer model (Liu et al., 2021) for masked image modeling (Xie et al., 2022) using the CIFAR10 dataset. Following the standard practice of this task, we select the model that gives the lowest validation loss among the last 50 epochs as the final output. The results are shown in Table 9. We can see that RAMDA attains the lowest validation loss and has a low-rank level almost identical to that of RMDA. On the other hand, ProxSGD and ProxGen have worse low-rank levels.

Table 8: Low-rank level and validation accuracy of different methods on training a six-layer fully-connected neural network with the FashionMNIST dataset for image classification.

| Algorithm | Validation accuracy | Low-rank level |
|---|---|---|
| MSGD | $89.95 \pm 0.29\%$ | - |
| ProxSGD | $87.54 \pm 0.52\%$ | $78.00 \pm 0.77\%$ |
| ProxGen | $86.66 \pm 0.33\%$ | $87.46 \pm 4.19\%$ |
| RMDA | $\mathbf{88.19 \pm 0.23\%}$ | $\mathbf{91.88 \pm 0.12\%}$ |
| RAMDA | $87.99 \pm 0.24\%$ | $89.59 \pm 0.42\%$ |

Table 9: Low-rank level and validation loss of different methods on pretraining a modified vision transformer model using the CIFAR10 dataset for masked image modeling.

| Algorithm | Validation loss | Low-rank level |
|---|---|---|
| AdamW | $0.0865 \pm 0.0001$ | - |
| ProxSGD | $0.1042 \pm 0.0003$ | $82.60 \pm 0.34\%$ |
| ProxGen | $0.1120 \pm 0.0019$ | $82.64 \pm 2.47\%$ |
| RMDA | $0.1054 \pm 0.0031$ | $\mathbf{86.23 \pm 0.41\%}$ |
| RAMDA | $\mathbf{0.1035 \pm 0.0016}$ | $\mathbf{86.20 \pm 0.35\%}$ |

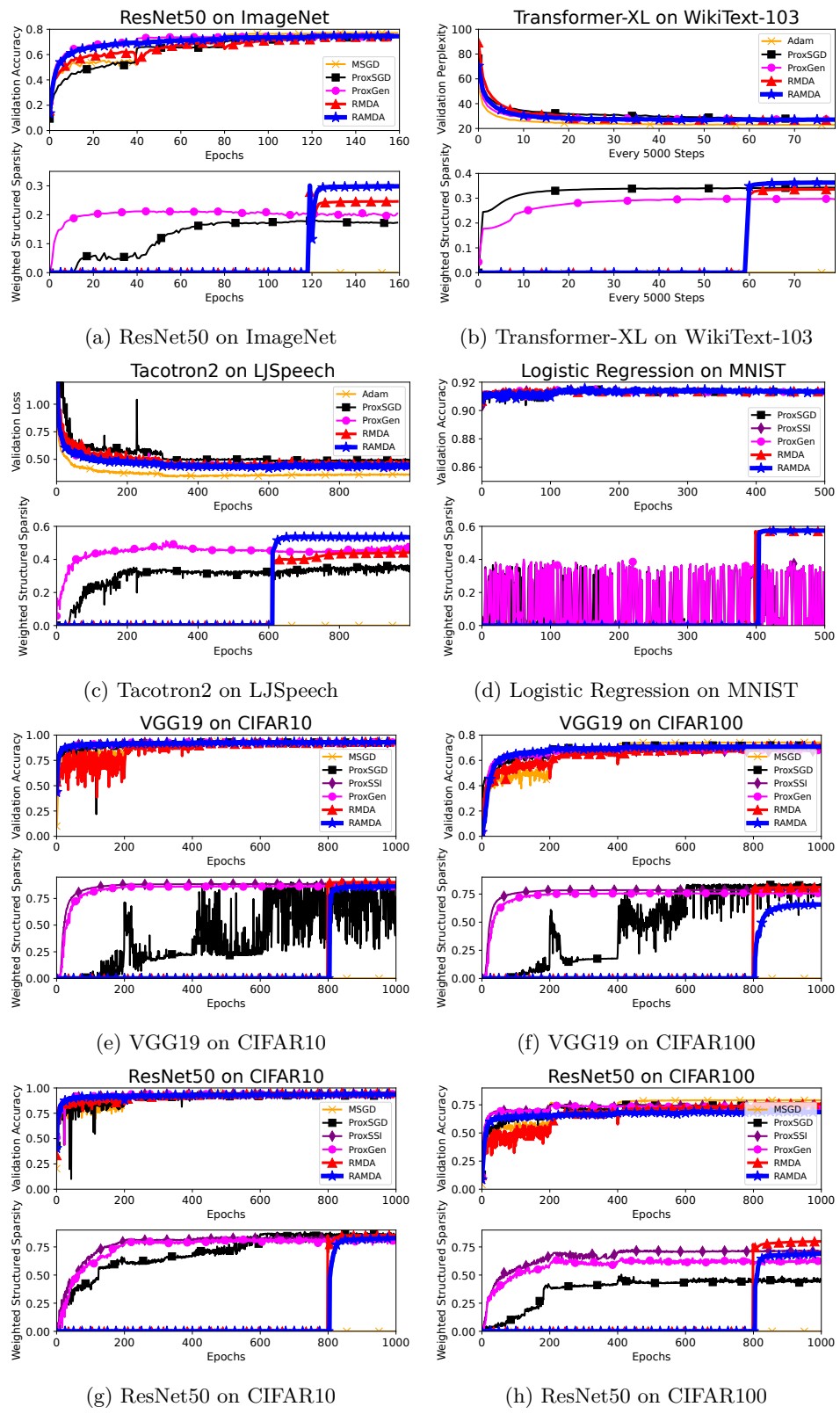

(a) ResNet50 on ImageNet

(b) Transformer-XL on WikiText-103

(c) Tacotron2 on LJSpeech

(d) Logistic Regression on MNIST

(e) VGG19 on CIFAR10

(f) VGG19 on CIFAR100

(g) ResNet50 on CIFAR10

(h) ResNet50 on CIFAR100

Figure 1: Group sparsity level and validation prediction performance v.s all epochs. In the plot for Transformer-XL, one step processes ten batches, and for our batch size of 64, one epoch consists of 8,401 batches.

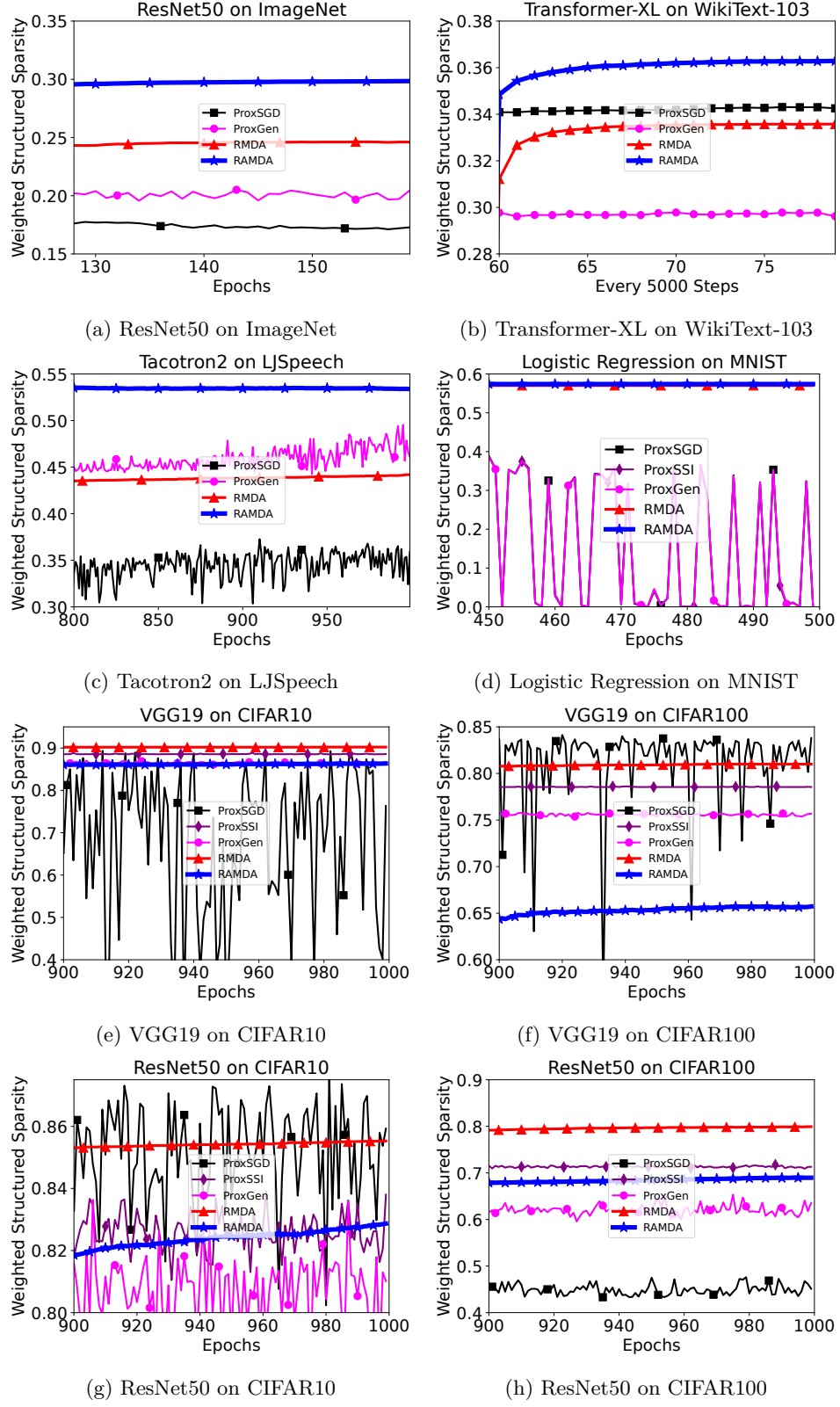

Figure 2: Group sparsity level at the last epochs.

