# OpenReview forum: "An Inexact Regularized Adaptive Algorithm with Manifold Identification for Training Structured Neural Networks"
_ICLR.cc/2024/Conference — Submitted to ICLR 2024_

### Official Review · Reviewer_N8rQ · 2023-10-30

**Soundness:** 3 good
**Presentation:** 3 good
**Contribution:** 3 good
**Rating:** 6
**Confidence:** 3

**Summary:**

The authors propose a variant algorithm of RMDA. Different from RDMA, the authors introduce a diagonal precondition matrix, based on the adaptive learning rate in the first-order algorithm. Further, to efficiently approximate the solution of the subproblem, the authors solve the subproblem with algorithm PG. Under some mild conditions, the authors provide convergence analysis of the proposed algorithm. The experimental results show the benefit of the proposed algorithm.

**Strengths:**

1. The authors introduce adaptive preconditioner, and inexact update of subproblem, while maintain the convergence of the algorithm.

2. The running time can be reduced 3x while the accuracy does not drop too much according to the experimental results.

**Weaknesses:**

Because I am not familiar with RDAM, I wonder about the difficulty of applying the adaptive preconditioner and inexact updates. Because extending SGD with such adaptive precondition is well-known, and applying inexact update in primal-dual update is well-known, I am not sure about the novelty of adding two techniques into RDAM.

**Questions:**

See weakness.

---

> ### Author Response · Authors · 2023-11-17
> **Reply**
>
> We thank the reviewer for the detailed evaluation of and the invaluable suggestions to our work.
> Our reply to the comments are as follows.
>
> 1. > Because I am not familiar with RDAM, I wonder about the difficulty of applying the adaptive preconditioner and inexact updates. Because extending SGD
> with such adaptive precondition is well-known, and applying inexact update in primal-dual update is well-known, I am not sure about the novelty of adding two techniques into RDAM.
>
> Indeed, adaptive methods in the case without a regularizer is well-studied and
> utilized in practice, and inexact updates in the mentioned primal-setting setting is also
> well-known.
> However, note that convergence of adaptive algorithms is not proven until very recently by [1], and its theoretical studies is not quite well studied.
> As mentioned in the paper, our algorithm can be seen as an extension of the MADGRAD algorithm of Defazio and Jelassi to the regularized setting, but their theoretical analysis covered the special case that the objective is convex only, while our analysis is for nonconvex problems, and we have the added critical component of manifold identification for obtaining structured models.
> The primal-dual updates the reviewer mentioned is also for convex problems (primal-dual relationship is meaningful only for convex problems, for otherwise the smallest possible duality gap could still be arbitrarily large).
>
> We also want to mention that even an extension seemingly very similar for different algorithms could be of significantly different difficulties.
> For example, we easily have convergence guarantees for both gradient descent and Newton's method with line search on strongly convex problems, and we can still get convergence guarantees for GD on nonconvex problems, but without modifications, classical Newton's method on nonconvex problems would easily fail to converge.
>
> More importantly, intuitively inexact updates could probably still guarantee
> asymptotic convergence of some first-order measure, but for manifold/structure
> identification it could be quite different.
> When the subproblems are solved exactly, first-order optimality condition in
> the subproblem ensures satisfaction of sufficient conditions for manifold identification,
> but when the subproblem is solved inexactly, the story is totally different.
> For demonstration, we will use the basic proximal gradient (non-stochastic)
> algorithm with a very simple strongly convex toy example of
>
> $$
> \min_{x \in R^2}\quad F(x) \coloneqq \frac12 (x - \hat x) ^\top (x-\hat x)  + \|\|x\|\|_1,\quad \hat x \coloneqq (0.3, 2).
> $$
>
> The only optimal solution is $x^* = (0, 1)$, and the gradient of the smooth (quadratic) part has Lipschitz constant $L = 1$.
> Consider that we start from $x_0 = 0$, then the subproblems are of the form
>
> $$
> \min_{x \in R^2}\quad (x_k - \hat x)^\top x + \frac12 x^\top x + \|\|x\|\|_1 = \min \quad F(x) + \text{constant}.
> $$
>
> We consider inexact solutions to the subproblem (and thus also to the original problem) of the form
> $$
> x_k = (f(k), 1 + f(k))
> $$
>  for any decreasing function $f$ that converges asymptotically to
> $0$ arbitrarily fast with $k$, but $f(k) \neq 0$ for any given value of $k$.
> Then clearly, $x_k$ converges to $x^*$ arbitrarily fast, but due to this
> inexactness, no matter how small, structure identification (here the structure
> is that the 1st coordinate becomes $0$) is never achieved.
> This shows that careful design of the inexactness condition and theoretical
> analysis is needed to ensure identification of the active manifold, which is of
> high importance for training structured models, as we have also seen in the
> experiments.
>
> We also note (as we have remarked in the paper before Theorem 4) that when
> applied to the existing framework of Yun et al. (2021), our inexactness
> condition leads to much stronger guarantees than that in the
> literature by Deleu and Bengio (2021), as their analysis with inexact solutions
> requires the regularizer to be smooth, which induces no structure, but ours
> applies to general nonsmooth regularizers.
> Moreover, our condition is easily checkable, but their inexact condition on the
> distance between the current subproblem objective value and the optimum is not
> verifiable because the optimum is not known a priori.
> This also shows even regarding convergence only, suitable inexactness is not
> well-studied yet.
>
> Overall, we indeed combined several known components to design our new
> algorithm, but the difficulty of combining multiple items is not
> just the sum of the individual items, just like one wouldn't say that since
> convex nonsmooth optimization is well-studied and nonconvex smooth optimization
> is well-studied, so nonconvex nonsmooth problems can also be solved easily.
> And indeed before this work, an adaptive regularized training algorithm that
> identifies the locally optimal structure is still missing, and our method
> indeed exhibits superior performance over existing methods over representative
> large-scale deep learning works.

---

> > ### Author Response · Authors · 2023-11-17
> > **Reference**
> >
> > **Reference**:
> >
> > [1] Zhang, Yushun, Congliang Chen, Naichen Shi, Ruoyu Sun, and Zhi-Quan Luo. "Adam can converge without any modification on update rules." Advances in Neural Information Processing Systems 35 (2022): 28386-28399.

---

### Official Review · Reviewer_a8Jk · 2023-10-31

**Soundness:** 4 excellent
**Presentation:** 4 excellent
**Contribution:** 3 good
**Rating:** 6
**Confidence:** 3

**Summary:**

The authors propose an inexact regularized adaptive dual averaging algorithm with momentum, which they call RAMDA, for training structured neural networks with the help of regularization. The authors provide a number of theoretical guarantees for their algorithm, such as: after a finite number of steps, the structure of the iterates of RAMDA are identical to the structure induced by the regularization at the stationary point. RAMDA also makes use of manifold identification by producing stochastic estimators of the gradient that almost surely converge to the true gradient. The authors also propose a general iterative subroutine for approximately solving a particular subproblem found in RAMDA and many existing frameworks. They also provide extensive numerical experiments, comparing their method RAMDA with other state-of-the-art methods.

**Strengths:**

- The method seems novel and interesting.
- The authors provide a stronger result than that of Deleu & Bengio (2021), showing that their subproblem solver for equations (4) and (6) can be effectively applied to the general framework of Yun et al. (2021) and maintaining the same convergence guarantees.
- The authors provide novel proofs that show the iterates of RAMDA (their method) possess the same structure as that at the point of convergence (Theorem 3).
- The authors provide extensive experiments that demonstrate the effectiveness of their method.

**Weaknesses:**

- The authors don't necessarily prove convergence of their algorithm to a point (please correct me if I'm wrong). They remark after Theorem 2, that proving this is difficult even for stochastic gradient descent. Could this be done for gradient descent? Perhaps one needs further assumptions as well.

- While the authors speak about structure in generality throughout the paper, the experiments only examine one type of structure (sparsity). It would be interesting to empirically examine other types of structure.

**Questions:**

- What is the reason for disabling weight decay in the experiments?
- Can RAMDA be applied to other structure types, besides sparsity?
- In Table 4, the text says RAMDA gives the best validation accuracy, but it's actually MSGD. Perhaps you mean that RAMDA gives the best validation accuracy out of the sparsity-inducing methods. Same point for the other tables.

---

> ### Author Response · Authors · 2023-11-17
> **Reply (1)**
>
> We would like to thank the reviewer for the detailed evaluation of and the
> invaluable suggestions to our work.
> Our reply to the comments are as follows.
>
>
> 1.
> > The authors don't necessarily prove convergence of their algorithm to a point (please correct me if I'm wrong). They remark after
>   Theorem 2, that proving this is difficult even for stochastic gradient descent. Could this be done for gradient descent? Perhaps one
>   needs further assumptions as well.
>
>   It is indeed true that we do not have guarantees for the iterates to converge.
>   However, when the loss is lower-bounded (by 0, as is usually the case for machine learning tasks) and the regularizer is coercive
>   (which is the case for the group-LASSO norm we used in our current experiments), the objective function is level-bounded, meaning
>   that when the objective values do not blow up, the iterates are also bounded.
>   In this case, at least a subsequence of the iterates converge to a limit point.
>   And surely if there are certain hyperparameter settings that make the objective
>   value blow up, they would have been excluded in the hyperparameter search, so our algorithm essentially has at least a limit point in
>   practical settings.
>
>   Even for (non-stochastic) gradient descent without a regularizer, convergence of the iterates would require additional assumptions.
>   For example, convexity would be one of such conditions [1].
>   For nonconvex problems, the most prominent condition for iterate convergence is the Kurdyka-Lojasiewicz
>   condition at a limit point or a compact set of limit points.
>   This condition means that the objective value is dominated by a polynomial of
>   the subgradient, in a neighborhood of the limit point(s). See, for example, the
>   canonical works [2,3,4].
>   However, those are for subgradient-like descent methods that monotonically
>   decrease the objective value every iteration, and they still need to make the
>   additional assumption that the iterates stay in a bounded area.
>
>   For dual-averaging methods this is even more difficult.
>   Up to our knowledge, the only existing works that have guarantees on
>   convergence of the iterates for dual-averaging-type methods need a global
>   convexity and a local strong-convexity condition. See [5,6].
>
>
> 2.
> > While the authors speak about structure in generality throughout the paper, the experiments only examine one type of structure (sparsity). It would be interesting to empirically examine other types of structure.
>
> > Can RAMDA be applied to other structure types, besides sparsity?
>
> As our theory shows, as long as the regularizer associated with the structure
> is partly smooth and prox-regular at the point of convergence, RAMDA can be
> applied to identify the active manifold. For convergence, we the
> subdifferential of the regularizer should be outer semicontinuous, as required
> in Theorem 2.
> Many popular regularizers satisfy these conditions (all convex regularizers are
> prox-regular everywhere and their subdifferential is outer semicontinuous
> everywhere), so
> RAMDA is not confined to sparsity or group sparsity, and we picked it
> simply because it seems to be the most widely considered structure in practice
> for a wide spectrum of machine learning problems including but not limited to
> deep learning.
> There are also some other regularizers with different structures like the
> L-infinity norm, nuclear norm, and the total-variation norm that are popular in
> various tasks in machine learning and image processing that are partly smooth.
> See, for example, Table 1 of [7] for some widely-used regularizers and their corresponding structures/manifolds.
>
> We have added a preliminary experiment of a layer-wise low-rank structure in Appendix D to show that our method indeed works on other structures widely used in machine learning as well.
>
>
> 3.
> > What is the reason for disabling weight decay in the experiments?
>
> One reason is that weight decay is disabled in the code provided by NVIDIA for training Transformer-XL with the Wikitext-103 data, see https://github.com/NVIDIA/DeepLearningExamples/blob/master/PyTorch/LanguageModeling/Transformer-XL/pytorch/wt103_base.yaml.
>
> Another reason is that weight decay is in spirit akin to L2 regularization for avoiding overfitting, and in our tasks, we already have another regularizer that will also provide such functionality, so adding weight decay probably would not affect the performance much but it would for sure increase the cost of hyperparameter tuning.
>
> But to be honest, the major reason for us is budget concerns.
> Adding in weight decay would lead to an additional hyperparameter for tuning,
> and the overall cost could easily become 5-10 times higher, while for our limited
> budget, we think that going for larger problems but with less hyperparameter
> tuning would be more representative and meaningful than going for smaller
> problems with an exhaustive hyperparameter search.

---

> > ### Author Response · Authors · 2023-11-17
> > **Reply (2)**
> >
> > 4.
> > > In Table 4, the text says RAMDA gives the best validation accuracy, but it's actually MSGD. Perhaps you mean that RAMDA gives the best validation accuracy out of the sparsity-inducing methods. Same point for the other tables.
> >
> > Indeed we meant for methods that induce structures, and sorry for the confusion.
> > We now have added double separation lines between the non-structural baseline and other structure-inducing training algorithms in Tables 4-6 in the revision to avoid confusion. We have also added a sentence in Sec 5 (after the itemization) stating that the baseline is for reference only and our comparison is for the methods that produce structured models.
> >
> >
> >
> >
> > **References:**
> >
> > [1] Burachik, Regina, L. M. Graña Drummond, Alfredo N. Iusem, and Bernar Fux Svaiter. "Full convergence of the steepest descent method with inexact line searches." Optimization 32, no. 2 (1995): 137-146.
> >
> > [2] Attouch, Hedy, Jérôme Bolte, and Benar Fux Svaiter. "Convergence of descent methods for semi-algebraic and tame problems: proximal algorithms, forward–backward splitting, and regularized Gauss–Seidel methods." Mathematical Programming 137, no. 1-2 (2013): 91-129.
> >
> > [3] Bolte, Jérôme, Aris Daniilidis, and Adrian Lewis. "The Łojasiewicz inequality for nonsmooth subanalytic functions with applications to subgradient dynamical systems." SIAM Journal on Optimization 17, no. 4 (2007): 1205-1223.
> >
> > [4] Bolte, Jérôme, Shoham Sabach, and Marc Teboulle. "Proximal alternating linearized minimization for nonconvex and nonsmooth problems." Mathematical Programming 146, no. 1-2 (2014): 459-494.
> >
> > [5] Lee, Sangkyun, and Stephen J. Wright. "Manifold Identification in Dual Averaging for Regularized Stochastic Online Learning." Journal of Machine Learning Research 13, no. 55 (2012): 1705-1744.
> >
> > [6] DUCHI, JOHN C., and FENG RUAN. "ASYMPTOTIC OPTIMALITY IN STOCHASTIC OPTIMIZATION." The Annals of Statistics 49, no. 1 (2021): 21-48.
> >
> > [7] Liang, Jingwei, Jalal Fadili, and Gabriel Peyré. "Local convergence properties of Douglas–Rachford and alternating direction method of multipliers." Journal of Optimization Theory and Applications 172 (2017): 874-913.

---

> > > ### Comment · Reviewer_a8Jk · 2023-11-20
> > >
> > > I thank the authors for their reply. I have also read the other reviews and the discussion that followed (as of this writing), as well as the updates to the paper.
> > >
> > > I am grateful that they included another experiment, albeit preliminary, to showcase a different type of structure (low-rankness). I do see that RMDA is the best performer, as opposed to the algorithm in the paper, RAMDA. But RAMDA and RMDA are similar, and they both outperform the other methods, ProxSGD and ProxGen.
> > >
> > > My questions and concerns have been answered, and as it currently stands I am inclined to keep my score.

---

> > > > ### Author Response · Authors · 2023-11-23
> > > > **A slight update**
> > > >
> > > > We would like to update that we have conducted another experiment of a larger scale and a more useful model for the low-rank experiment.
> > > > We now also report a result of pretraining a low-rank modified version of vision transformer on CIFAR10 using the masked image modeling framework with a nuclear-norm regularizer. In this result, RAMDA achieves the lowest validation loss, and the low-rank level of RAMDA and RMDA are almost identical and better than other methods.

---

### Official Review · Reviewer_VVRX · 2023-11-05

**Soundness:** 2 fair
**Presentation:** 1 poor
**Contribution:** 2 fair
**Rating:** 5
**Confidence:** 3

**Summary:**

The paper presents a new algorithm named Regularized Adaptive Dual Averaging with Momentum (RAMDA) for training structured neural networks. The main problem that this paper tackles is that optimization algorithms only guarantee a presence of structure like sparsity at the convergence point but don’t offer any guarantee of the structure at points close to the stationary points which is what we use in practice. Utilizing the theory of manifold identification, RAMDA ensures that after a finite number of steps, the structures of the iterates are identical to the structure at the stationary point of convergence and their algorithm is adaptive. To achieve this, the authors tackle the challenges of solving subproblems in the presence of preconditioners and regularization terms by proposing an iterative subroutine that approximately solves these issues, while still maintaining convergence guarantees. The paper includes experiments with modern neural networks in computer vision, language processing, and speech tasks, demonstrating that RAMDA achieves better structured sparsity and prediction performance than the state of the art.

**Strengths:**

- The problem of guaranteeing structure at close to the stationary points seems important which is not very well studied.

**Weaknesses:**

- The paper is not well written and it is very hard to understand the exact comparison with existing approaches and what the paper is trying to achieve. I had to go and read the previous RMDA paper to understand what the problem is.
- Moreover, there are a lot of statements in the paper which seems like is the contribution of the paper but are actually taken from previous works. This problem of guaranteeing structure near the stationary point has already been studied. Like the challenges in variance reduction with data augmentation have been already dealt with in the RMDA paper. The main contribution in this work is of making the RMDA algorithm adaptive. Moreover, there are other works as the authors mention like Defazio and Jelassi which had already combined adaptiveness and momentum. It is not clear how does this work differs from those previous works.

**Questions:**

- On page 2, the authors mention the active manifold of being the lowest rank. I am not sure I understand this statement and how it fits in with the rest of the paper. Do the authors show that the manifold that they identify is optimal? Was it also shown in the RMDA work?
- How does the computational efficiency of RAMDA compare to other state-of-the-art methods, particularly in training very large neural networks?
- In the experiments conducted, were there any scenarios where RAMDA did not perform as well as expected?
- For the vision tasks, why is the performance not compared to other algorithms?
- In the results section, the performance of RAMDA is better than RMDA. Why is that? Because, it seems like RAMDA is achieving the same structure just can achieve it in fewer iterations because of the adaptive component.

---

> ### Author Response · Authors · 2023-11-10
> **Request for clarification of one of the questions**
>
> We would like to thank the reviewer for the careful read of our manuscript and the review.
> Before preparing for a full reply, we would like to first request a clarification.
> The reviewer asked
>
> > For the vision tasks, why is the performance not compared to other algorithms?
>
> But in section 5.2, we did compare all the algorithms listed in Table 4. Do you mean Tables 2-3 in section 5.1?

---

> ### Author Response · Authors · 2023-11-17
> **Reply (1)**
>
> We would like to thank the reviewer for the detailed evaluation of and the
> invaluable suggestions to our work.
> Our reply to the comments are as follows.
>
> 1. Regarding the comments in the weakness part, we have modified the
> introduction to clarify the problem we tackle and what are the main focus and
> contributions of this paper.
> The reviewer also mentioned that
>
> > This problem of guaranteeing structure near the stationary point has already been studied. Like the challenges in variance reduction with data augmentation have been already dealt with in the RMDA paper.
>
> It is true that guaranteeing structure near the stationary point has been
> studied, as we have used existing tools of manifold identification from
> nonlinear optimization. The RMDA paper has also studied this, but that is for a
> different algorithm, and without inexactness.
> With inexactness, analysis becomes more difficult, and with adaptiveness added,
> the algorithm is totally different and convergence guarantees and
> identification guarantees are nontrivial.
> For example, convergence guarantees of adam is not established until last year
> by Zhang et al [1], and convergence guarantees of adagrad are also not proven until
> very recently (the original adagrad paper by Duchi et al dealt with convex
> problems only, and the analysis is for regret guarantees for online settings
> but not for the traditional convergence guarantees),
> while convergence of SGD has been known for decades.
> On the other hand, for identification guarantees, the major difficulties are
> from the inexactness in the subproblem solving, and inexactness emerges only
> when we combine adaptiveness and regularization.
> When the subproblems are solved exactly, first-order optimality condition in
> the subproblem can easily guarantee sufficient conditions for manifold
> identification, but when the subproblem is solved inexactly, the story is
> totally different and thus difficult.
> For demonstration, we will use the basic proximal gradient (non-stochastic)
> algorithm with a very simple strongly convex toy example of
>
> $$
> \min_{x \in R^2}\quad F(x) \coloneqq \frac12 (x - \hat x) ^\top (x-\hat x)  + \|\|x\|\|_1,\quad \hat x \coloneqq (0.3, 2).
> $$
>
> The only optimal solution is $x^* = (0, 1)$, and the gradient of the smooth (quadratic) part has Lipschitz constant $L = 1$.
> Consider that we start from $x_0 = 0$, then the subproblems are of the form
>
> $$
> \min_{x \in R^2}\quad (x_k - \hat x)^\top x + \frac12 x^\top x + \|\|x\|\|_1 = \min \quad F(x) + \text{constant}.
> $$
>
> We consider inexact solutions to the subproblem (and thus also to the original problem) of the form
> $$
> x_k = (f(k), 1 + f(k))
> $$
>  for any decreasing function $f$ that converges asymptotically to
> $0$ arbitrarily fast with $k$, but $f(k) \neq 0$ for any given value of $k$.
> Then clearly, $x_k$ converges to $x^*$ arbitrarily fast, but due to this
> inexactness, no matter how small, structure identification (here the structure
> is that the 1st coordinate becomes $0$) is never achieved.
>
>
> Our goal is to combine adaptiveness, regularization, momentum, and structure
> identification (which requires variance reduction), for it is known that
> adaptive methods are superior for many deep learning models, but combination of
> these elements result in new challenges including those we illustrated above
> and beyond.
>
>
> > The main contribution in this work is of making the RMDA algorithm adaptive. Moreover, there are other works as the authors mention like Defazio and Jelassi which had already combined adaptiveness and momentum. It is not clear how does this work differs from those previous works.
>
> If it is just about combining adaptiveness and momentum, it has already been done in Adam.
> Whether variance reduction is achieved or not after adaptiveness is added to dual
> averaging has not been studied by Defazio & Jelassi, and their convergence
> analysis is for convex problems only, but our analysis is for nonconvex problems.
> Moreover, the key difference between our work and theirs is that we have a
> regularizer in the objective function to promote structures, and this results
> in inexact updates, while their algorithm for smooth optimization does not
> involve any inexactness.
> Our key result is that with inexactness and a preconditioner for adaptiveness,
> we still ensured manifold identification.
> Adaptiveness in training structured models is still missing and it is crucial
> for getting good models, as we have seen in our experiments for modern large
> deep learning models.
> We have updated our introduction to clarify our contributions and differences
> with existing works.

---

> > ### Author Response · Authors · 2023-11-17
> > **Reply (2)**
> >
> > 2.
> > > On page 2, the authors mention the active manifold of being the lowest rank. I am not sure I understand this statement and how it fits in with the rest of the paper. Do the authors show that the manifold that they identify is optimal? Was it also shown in the RMDA work?
> >
> > Take the sparsity experiment as an example, each sparsity pattern forms a
> > manifold (here just a linear subspace), and the collection of all possible
> > sparsity patterns form the collection of possible manifolds associated with the
> > regularizer.
> > If we have a sequence of points $\{x^k\}$ converging to a certain point $x^*$
> > with a certain sparsity pattern, eventually $x^k$ cannot contain more zero
> > elements than $x^*$ for $k$ large.
> > For example, if $x^k_i = 0$ for all $k$ large, than clearly $\|x^k - x^*\| \geq |x^*_i| > 0$ and $\{x^k\}$ does not converge to $x^*$.
> > But on the other hand, as we have seen in the example above, $\{x^k\}$ could be all in a super-manifold (a subspace with a lower sparsity level) of the active manifold and still converge to $x^*$.
> > The same argument holds for other manifolds such that we cannot have a sequence inside a lower-dimensional manifold converging to a point with an active manifold of a higher dimension.
> > Thus the active manifold is optimal in the local sense for all sequences converging to the same point.
> > Our Theorem 2 therefore shows this kind of optimality.
> > Note that we do not make any claim on the structure being globally optimal, which is NP-hard to determine.
> >
> > This argument is actually from Appendix B.1 of that RMDA paper, and we thought
> > that we do not need to repeat that same argument in our work, so we did not
> > make further arguments.
> > Now we have revised our introduction to make a similar argument to clarify the
> > concept and implication of this low-rankness and locally optimal structure.
> >
> >
> > 3.
> > > How does the computational efficiency of RAMDA compare to other state-of-the-art methods, particularly in training very large neural networks?
> >
> > We have updated our Tables 5 and 6 to include the computational efficiency comparison between RAMDA and other methods (all algorithms were run for the same number of epochs, so average time per epoch comparison is equivalent to total time comparison) on the larger models of Transformer-XL and Tacotron2.
> > We can see that RAMDA and Proxgen are indeed slightly slower than other methods due to the subproblem routine, but the time is not increased much (around 11% on Transformer-XL and less than 3% for RAMDA, and even less for Proxgen with our subproblem solver).
> > The comparison in Table 3 also shows how much more efficient our proposed subproblem solver is than the state of the art.
> >
> >
> > 4.
> > > In the experiments conducted, were there any scenarios where RAMDA did not perform as well as expected?
> >
> > As mentioned in our section 5.2, there are cases that RAMDA performs not the
> > best for some smaller scale image classification problems, as we have shown in
> > Table 7 of Appendix B.
> > But we would say that this is quite expected, as have been shown by [2], that
> > non-adaptive methods tend to perform worse on computer vision tasks.
> > A similar observation on CIFAR datasets was also made for the official
> > implementation of MADGRAD (Defazio and Jelassi, 2022), see
> > https://github.com/facebookresearch/madgrad/issues/14
> >
> > We do not claim that RAMDA is always the best method, especially since there are
> > tasks on which adaptive methods are known to be unfavorable.
> > But for tasks that adaptiveness is widely used, we observe that RAMDA tends to
> > perform quite well as our large-scale experiments have shown.

---

> > > ### Author Response · Authors · 2023-11-17
> > > **Reply (3)**
> > >
> > > 5.
> > > > For the vision tasks, why is the performance not compared to other algorithms?
> > >
> > > For the Imagenet problem in Section 5.2, we already have comparison with other
> > > algorithms, so we assume that you are asking about Tables 2-3 in Section 5.1.
> > > The purpose of this subsection is to compare subproblem solvers (Table 3) and
> > > settings (Table 2) in terms of efficiency and how the inexactness affects model
> > > performance, in order to justify that the proposed subproblem solver and
> > > inexact criterion are efficient and suitable.
> > > We therefore only considered those involving our subproblem solver.
> > > The purpose of Table 2 is to ensure that our early stopping criterion does not
> > > affect the final model much so that we can apply it as the default setting,
> > > while the purpose of Table 3 is to compare our PG solver and the Newton-Raphson
> > > solver proposed by Deleu and Bengio.
> > >
> > > Also note that in Table 7 in Appendix B, we actually have already presented the
> > > performance comparison with other algorithms of exactly those tasks in Tables
> > > 2-3.
> > > (There was a typo in the original version of Table 7: the last part should be results on CIFAR100 instead of CIFAR10.)
> > > We put those in the appendix instead of the main paper because:
> > > 	1. We are out of space in the main paper
> > > 	2. These results are less important because they are of smaller models and smaller datasets, while we care more about results on large-scale deep learning tasks. Comparisons with existing methods in the main paper are all conducted in rather large-scale representative tasks.
> > > 	3. Similar to other adaptive algorithms, we do not expect RAMDA to be the best algorithm for smaller scale vision tasks, as shown by [2], and our focus is indeed tasks beyond computer vision.
> > >
> > >
> > > 6.
> > > > In the results section, the performance of RAMDA is better than RMDA. Why is that? Because, it seems like RAMDA is achieving the same structure just can achieve it in fewer iterations because of the adaptive component.
> > >
> > > We also had similar concerns about difference in the convergence speed, and thus for all the three tasks, we ran all algorithms for sufficiently long epochs, actually slightly longer than the usual setting, to make sure that the non-adaptive algorithms also have already converged.
> > > The particular settings are available in the experimental codes in the supplementary materials, and we summarize the settings here.
> > >
> > > 	Resnet50/Imagenet: For all algorithms:
> > > 	We run: 160 epochs
> > > 	Usual setting: 90 epochs (https://github.com/pytorch/vision/tree/main/references/classification#image-classification-reference-training-scripts)
> > > 	90 epochs is also the setting described in Appendix A of Defazio & Jelassi (2022).
> > >
> > > 	Transformer-XL/WikiText-103: For all algorithms:
> > > 	We: 400,000 steps
> > > 	Original authors of Transformer-XL: 200,000 steps (https://github.com/kimiyoung/transformer-xl/blob/master/pytorch/run_wt103_base.sh#L20)
> > >
> > > 	We could not find a suitable reference for the default setting for Tacotron2, but we also made sure that the number of epochs are long enough for al methods to converge.
> > >
> > > Convergence plots are also already given in Appendix C in the last few pages in our initial submission. We can see that the validation accuracy/perplexity of all algorithms have already become stable at the late stage, so the performance superiority is clearly not related to convergence speed difference.
> > >
> > > Now, let us go back to the first half of this question. The major reason of the superiority of RAMDA over RMDA is probably because adaptive algorithms are favorable for these tasks, which has been quite widely known in the literature. Adaptiveness, especially at the early stage, guides the iterates to a different region and thus the iterates produced by adaptive algorithms converge to a point different from the point of convergence of their nonadaptive counterparts. This is also why in the usual non-regularized setting of these models/datasets, Adam but not MSGD is almost always used as the default training algorithm by most researchers and practitioners.
> > > This was also our original motivation: training of these models clearly would
> > > benefit from adaptive algorithms, but there was no adaptive algorithms that
> > > stably identifies the locally optimal structure.
> > > The numerical results also show that indeed adaptive methods are needed for those tasks.
> > >
> > >
> > > **References**:
> > >
> > > [1] Zhang, Yushun, Congliang Chen, Naichen Shi, Ruoyu Sun, and Zhi-Quan Luo. "Adam can converge without any modification on update rules." Advances in Neural Information Processing Systems 35 (2022): 28386-28399.
> > >
> > > [2] Wilson, Ashia C., Rebecca Roelofs, Mitchell Stern, Nati Srebro, and Benjamin Recht. "The marginal value of adaptive gradient methods in machine learning." Advances in neural information processing systems 30 (2017).

---

### Author Response · Authors · 2023-11-17
**Summary of updates**

We thank all the reviewers for their careful evaluation of the manuscript and
the invaluable comments and suggestions that led to the improvements in
our revision.
Here we summarize our major changes:

1. **Clarification of contributions and difference with existing works**

Following the questions and concerns of reviewers VVRX and N8rQ, we have
revised our introduction to clarify our contributions, the differences with existing
works, and what are the challenges and how we tackled them.
Specifically, adaptiveness, inexactness, variance reduction, and
proximal/regularization are all components that have been studied in the
literature, but it does not mean that their combination is also easy.
This is particularly the case for deep learning because empirical performance
of an algorithm is equally important as its theoretical guarantees for deep
learning problems.
What we have achieved is having adaptiveness and manifold identification in the
regularized setting simultaneously, with an efficient subproblem solver and
theoretical guarantees as well as extensive numerical results to validate its
superiority in training neural network models, especially for architectures and
tasks for which adaptive methods are widely-known to outperform their nonadaptive
counterparts.

2. **More experimental results** (Sec 5):

To answer the question of reviewer VVRX, we now show the running time per
epoch (for the same problem, all training algorithms are run for the same
number of epochs, so this is equivalent to the total running time) of different
training algorithms in Sections 5.3 and 5.4 to verify that indeed our algorithm
has a running cost comparable to existing methods.
Following the suggestion of reviewer a8Jk, we are also running an experiment
for a different structure, and will report the results later when it is done.

---

> ### Author Response · Authors · 2023-11-19
> **A new experiment**
>
> In the latest revision, we have included a preliminary experiment of training a neural network with low-rank structures at each layer. in Appendix D. Low-rankness is selected because we feel it is also one of the most widely used structures.
> We treat each layer as a matrix and use the nuclear norm to induce low-rank solutions. The result shows that our method indeed works on other structures popular in machine learning.

---

### Meta-Review · Area_Chair_DhiN · 2023-12-18

**Metareview:**

This paper proposes an inexact regularized adaptive dual averaging algorithm with momentum for solving regularized expectation minimization problems and studies its convergence behavior using the theory of manifold identification. As noted by several reviewers, the paper combines and extends various existing results in the literature. There should be a more thorough discussion of these related works, so as to elucidate the contributions of this paper. Furthermore, the various conditions and assumptions in Theorem 3, which are formulated in the language of variational analysis, should be better connected to the machine learning applications of interest. Based on the above and the borderline evaluation of the reviewers, I regrettably cannot recommend acceptance of the paper at this point.

**Justification For Why Not Higher Score:**

The contributions of the paper do not appear to be substantial in view of existing works.

**Justification For Why Not Lower Score:**

N/A

---

### Decision · Program_Chairs · 2024-01-16

Reject